# DeepRTL: Bridging Verilog Understanding and Generation with a Unified Representation Model

**Yi Liu[1,2], Changran Xu[1,2], Yunhao Zhou[1,2], Zeju Li[1,2], Qiang Xu[1,2]**
[1]The Chinese University of Hong Kong    [2]National Technology Innovation Center for EDA
`{yliu22,zjli24,qxu}@cse.cuhk.edu.hk`
`{xxuchangran,yunhaoz.cs}@gmail.com`

## Abstract

Recent advancements in large language models (LLMs) have shown significant potential for automating hardware description language (HDL) code generation from high-level natural language instructions. While fine-tuning has improved LLMs' performance in hardware design tasks, prior efforts have largely focused on Verilog generation, overlooking the equally critical task of Verilog understanding. Furthermore, existing models suffer from weak alignment between natural language descriptions and Verilog code, hindering the generation of high-quality, synthesizable designs. To address these issues, we present DeepRTL, a unified representation model that excels in both Verilog understanding and generation. Based on CodeT5+, DeepRTL is fine-tuned on a comprehensive dataset that aligns Verilog code with rich, multi-level natural language descriptions. We also introduce the first benchmark for Verilog understanding and take the initiative to apply embedding similarity and GPT Score to evaluate the models' understanding capabilities. These metrics capture semantic similarity more accurately than traditional methods like BLEU and ROUGE, which are limited to surface-level n-gram overlaps. By adapting curriculum learning to train DeepRTL, we enable it to significantly outperform GPT-4 in Verilog understanding tasks, while achieving performance on par with OpenAI's o1-preview model in Verilog generation tasks. Our code and datasets are available at https://github.com/PeterLau61/DeepRTL.

## 1 Introduction

The development of powerful large language models (LLMs), such as OpenAI's GPT-4 (Achiam et al., 2023), has brought transformative benefits to diverse fields (Wei et al., 2024; Jin et al., 2023), including electronic design automation (EDA). These models help streamline the hardware design process by generating hardware description language (HDL) code, like Verilog, from user-defined specifications (Pearce et al., 2020; Chang et al., 2023). The use of LLMs in EDA has opened new avenues for agile chip design, wherein hardware designers can specify requirements through natural language prompts, potentially boosting both creativity and efficiency in chip design processes.

Despite the adaptability of commercial LLMs like GPT-4 in the EDA domain, the proprietary nature of many designs necessitates the development of a tailored model, which requires fine-tuning a specialized model to ensure data security and customization for specific needs. Recent studies have attempted fine-tuning open-source LLMs for Verilog generation, demonstrating great potential for automated generation of Verilog code from high-level prompts (Thakur et al., 2024; Chang et al., 2024b; Zhang et al., 2024). However, these efforts often focus solely on Verilog generation, neglecting the equally critical task of Verilog understanding, *i.e.*, summarizing high-level functionality from Verilog code snippets using natural language. This capability is essential for effective communication among hardware designers, as it helps decipher complex code written by others, facilitating collaboration and comprehension. Moreover, even for Verilog generation, these works fail to establish a strong alignment between natural language and Verilog code, which potentially harms the models' performance. For example, Thakur et al. (2024) do not incorporate paired data of natural language and Verilog code in their dataset. Chang et al. (2024b) propose mapping the

Verilog Abstract Syntax Tree (AST) directly to natural language, but this approach is limited to line-level translation and produces descriptions that lack high-level semantics. To further bridge this gap, Zhang et al. (2024) introduce the MG-Verilog dataset with multi-level descriptions alongside corresponding code samples, but its small size and reliance on LLaMA2-70B-Chat for annotations raise quality concerns. Such poor alignment between natural language and Verilog code can degrade the generation performance, leading to generation of non-synthesizable or non-functional Verilog code.

To address these challenges, we introduce DeepRTL, a unified representation model that bridges Verilog understanding and generation. Achieving this requires an extensive collection of high-quality, hardware-specific datasets, which are scarce in the open-source community. To this end, we have meticulously curated a comprehensive Verilog dataset that ensures strong alignment between Verilog code and natural language across multiple levels. This dataset includes both open-source and proprietary Verilog design data. For the open-source data, we adopt the chain-of-thought (CoT) approach and use GPT-4, the most advanced model available, to generate precise natural language descriptions of the Verilog code. Human evaluations have verified that these annotations are approximately $90\%$ accurate, underscoring the dataset's high quality and reliability for training. For the proprietary data, we engage a team of professional hardware designers to provide detailed annotations, which capture intricate design features and further boost the dataset's quality. This comprehensive dataset enables us to develop DeepRTL capable of both understanding and generating Verilog code. By integrating high-quality annotations, the model enhances efficiency and accuracy in various design tasks.

We are the first to integrate the task of Verilog understanding into our model, addressing a significant gap left by previous works that focus exclusively on Verilog generation. These earlier efforts lack benchmarks to evaluate LLMs' understanding capabilities of Verilog code, prompting us to introduce the first benchmark for Verilog understanding. Our benchmark comprises one hundred diverse, high-quality Verilog designs and we have collaborated with professional engineers to develop precise high-level functional descriptions, which have been meticulously cross-checked by multiple designers to ensure their accuracy. In the software domain, traditional metrics like BLEU (Papineni et al., 2002) and ROUGE (Lin, 2004) scores have been commonly used to assess the similarity between generated code summaries and ground truth annotations (Wang et al., 2023a). However, these metrics focus primarily on lexical overlap and often fail to capture the true semantic meaning of the descriptions. To tackle this issue, we take the initiative to apply embedding similarity and GPT score for evaluation, both of which assess semantic similarity more effectively. Embedding similarity utilizes vector representations for semantic alignment, while GPT score uses advanced LLMs to assess the semantic coherence between descriptions. These metrics provide a more accurate means of evaluating generated descriptions against the ground truth annotations.

In our work, we employ CodeT5+ (Wang et al., 2023a), a family of encoder-decoder code foundation LLMs pre-trained on extensive software code, as the foundation to fine-tune on our dataset. Since the dataset includes hierarchical code summaries across multiple levels—line, block, and module, providing both detailed and high-level functional descriptions—we adapt curriculum learning for training. This begins with fine-tuning on line-level and block-level data, subsequently advancing to more complex module-level content. Such a structured approach enables the model to incrementally build foundational knowledge, which significantly enhances its performance in both Verilog understanding and generation tasks. In Verilog understanding, DeepRTL significantly outperforms GPT-4 across all metrics on the newly established understanding benchmark. In Verilog generation, it achieves comparable performance to OpenAI's o1-preview, the latest model designed for complex reasoning tasks including programming on the latest generation benchmark by Chang et al. (2024a).

## 2 RELATED WORKS

**Register Transfer Level in EDA.** Register Transfer Level (RTL) is a critical abstraction in EDA that describes the flow of data between registers and the logical operations on that data. This level is typically expressed using HDLs, with Verilog being the most widely used HDL in the industry. Consequently, the terms HDL and Verilog are used interchangeably in this work. In modern hardware design, engineers begin with specifications in natural language, which are then manually translated into HDLs before synthesizing circuit elements (Blocklove et al., 2023). This manual translation process is prone to human errors, leading to potential design flaws and inefficiencies. Automating this translation can significantly reduce errors and streamline the design process. Recent develop-

ments in artificial intelligence (AI) have enabled machine-based end-to-end translations, making this automation possible. And the ability to understand and generate Verilog code is crucial for advancing this automation in hardware design. For an introduction of Verilog, please refer to Appendix A.

**LLMs for EDA.** Recent advancements in LLMs have significantly impacted EDA, marking a transformative shift in hardware design (Chen et al., 2024). Researchers have examined the utilization of LLMs for Verilog code generation, with benchmarking results presented in Thakur et al. (2023); Liu et al. (2023b); Lu et al. (2024) showing the potential of these models to mitigate the design challenges faced by hardware developers. Furthermore, significant achievements have been achieved in fine-tuning for Verilog code generation (Chang et al., 2024b; Thakur et al., 2024), general RTL generation (Blocklove et al., 2023), and EDA tool script generation (Liu et al., 2023a; Wu et al., 2024). By reducing the need for extensive expertise in specific hardware, LLMs enable hardware developers to quickly design intricate hardware systems (Fu et al., 2023).

**Fine-tuning LLMs for Verilog Generation.** Despite the great potential of state-of-the-art (SOTA) LLMs, *e.g.*, OpenAI's GPT-4 (Achiam et al., 2023), in generating Verilog code, relying solely on them is insufficient due to the proprietary nature of hardware design. Besides, they are still limited in their ability to generate practical hardware designs (Fu et al., 2023). To address these limitations, recent studies have fine-tuned open-source LLMs on curated hardware design datasets (Liu et al., 2023b; Chang et al., 2024b; Thakur et al., 2024; Zhang et al., 2024), which has been shown to improve LLMs' performance in generating Verilog code. However, effective use of LLMs in hardware design requires high-quality, domain-specific data. Unfortunately, existing publicly available hardware datasets are often limited in size, complexity, or detail, hindering the effectiveness of LLMs in hardware design tasks. For example, datasets used in Thakur et al. (2023); Lu et al. (2024) contain fewer than 200 data points, making them suitable only for benchmarking rather than effectively fine-tuning. Meanwhile, other datasets, such as those employed in Liu et al. (2023b); Thakur et al. (2024), are overly simplistic, which hinder effective fine-tuning of LLMs. To improve alignment between natural language and Verilog code, Chang et al. (2024b) translate Verilog files to an AST and then map nodes to natural language with a predefined template. However, this method is limited to line-level Verilog code and the template-based descriptions lack semantic information. Furthermore, the MG-Verilog dataset (Zhang et al., 2024), despite featuring multi-level descriptions alongside code samples, is limited in size and its reliance on LLaMA2-70B-Chat for annotations raises quality concerns about the dataset. These alignment issues may hinder the fine-tuned LLMs' performance, leading to the generation of non-synthesizable or non-functional hardware source code. To address the limitations of previous studies, we introduce a novel high-quality dataset that aligns natural language with Verilog code at multiple levels: line, block, and module. It includes both detailed and high-level descriptions, integrating open-source and proprietary code to enhance its diversity and applicability. Unlike prior efforts focused solely on Verilog generation, we are the first to consider the crucial task of Verilog understanding. This comprehensive dataset enables the development of a unified representation model, DeepRTL, which excels in both Verilog understanding and generation, paving the way for significant advancements in hardware design automation.

**Curriculum Learning.** Curriculum learning is a training strategy inspired by human learning, where models are exposed to simpler tasks before advancing to more complex ones. This approach has been shown to accelerate convergence and improve model performance, particularly for tasks with hierarchical difficulty levels. Initially introduced in Bengio et al. (2009), curriculum learning has been applied to various domains, including natural language processing (Xu et al., 2020), computer vision (Wang et al., 2023b), and reinforcement learning (Narvekar et al., 2020). Recent work has demonstrated its efficacy in fine-tuning LLMs, where progressively increasing task complexity helps the models better capture intricate patterns (Campos, 2021). Notably, Na et al. (2024) apply curriculum learning to code language models, achieving significant improvements in the accuracy of code execution tasks. In this work, we adapt curriculum learning to train DeepRTL, utilizing our structured dataset with descriptions at varying levels of detail. This approach significantly enhances the model's capabilities in both Verilog understanding and generation.

## 3 DATASET AND UNDERSTANDING BENCHMARK

In this section, we introduce our dataset designed to enhance Verilog understanding and generation, which aligns natural language with Verilog code across line, block, and module levels with detailed

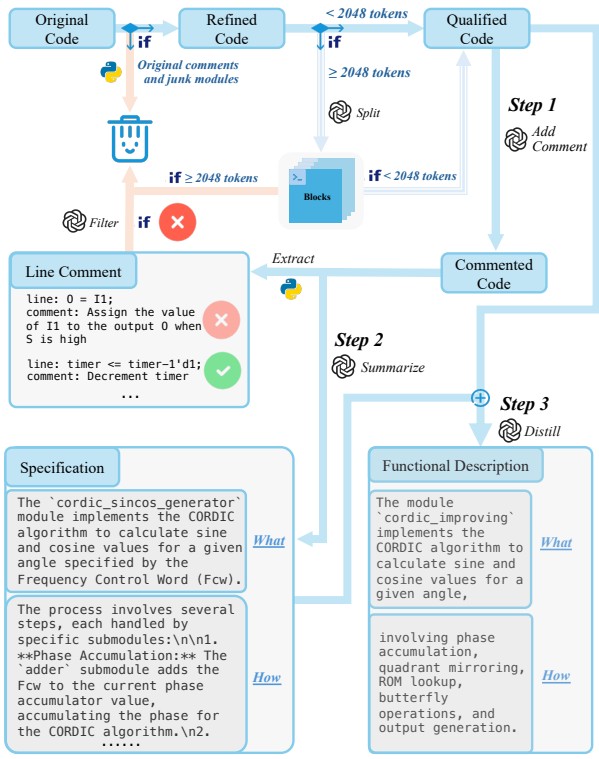

Figure 1: The overview of the data annotation process. We employ the CoT approach and the SOTA LLM, GPT-4, for annotation. Annotations span three levels—line, block, and module—providing both detailed specifications and high-level functional descriptions.

and high-level descriptions. By integrating both open-source and proprietary code, the dataset offers a diverse and robust collection that spans a broad spectrum of hardware design complexities. We employ GPT-4 along with the CoT approach for annotation, achieving about $90\%$ accuracy in human evaluations, confirming the dataset's high quality. We also introduce the first benchmark for Verilog understanding, setting a new standard for evaluating LLMs' capabilities in interpreting Verilog code.

## 3.1 DATASET SOURCE

Our dataset comprises both open-source and proprietary Verilog code. For the open-source part, we gather `.v` files from GitHub repositories using the keyword `Verilog`. These files are segmented into individual modules, each representing a distinct functional unit within the Verilog design. This segmentation is crucial given the limited context length of current LLMs, improving the efficiency and accuracy of the subsequent annotation and fine-tuning processes. We employ MinHash and Jaccard similarity metrics (Yan et al., 2017) to deduplicate these modules and exclude those predominantly made up of comments or lacking complete `module` and `endmodule` structures. Finally, this process results in a total of 61,755 distinct Verilog modules. For the proprietary portion, we incorporate a set of purchased intellectual properties (IPs) that enhance the variety and functional diversity of our dataset. This component includes a total of 213 high-quality, industry-standard Verilog modules. These IPs not only offer a range of advanced functions but also provide unique insights that complement the open-source data. Integrating these elements ensures a comprehensive dataset that captures a wide spectrum of hardware design practices.

## 3.2 DATASET ANNOTATION

We employ different annotation strategies for open-source and proprietary code. For open-source code, we utilize the CoT approach with the SOTA LLM, GPT-4, to provide annotations at multiple levels. As illustrated in Figure 1, we initially remove all comments from the original Verilog code

(resulting in refined code) to avoid training complications from incorrect or misleading comments. If the token count of a complete module exceeds 2048, the maximum context length for CodeT5+, we utilize GPT-4 to segment the module into smaller, manageable blocks such as `always` blocks. If the resulting blocks still exceed 2048 tokens, we will discard them. For modules and blocks with a token count below 2048 (qualified code), we then use GPT-4 to add informative comments, resulting in commented code (Step 1). From this commented code, we can extract line-level descriptions (pairings of single lines of code with natural language descriptions). To guarantee the accuracy and relevance of the inline comments, we use GPT-4o-mini to rigorously check each comment, ensuring that all line-level descriptions are strictly confined to the context of their respective lines without incorporating any extraneous or irrelevant information. For example, consider the line `"O = I1;"` annotated with `"Assign the value of I1 to the output O when S is high."`. Since we cannot deduce from this single line that `O` is the output and `S` is related, such descriptions are deemed inaccurate and are consequently excluded from the dataset to maintain training effectiveness. In Step 2, we use GPT-4 to generate a detailed specification for the commented code from Step1. This specification includes two main components: a summary of the code's functionality (what it does) and a comprehensive explanation of the implementation process (how it works). Finally, in Step 3, we combine the qualified code from Step 1 with the detailed specification generated in Step 2 to create high-level functional descriptions. To ensure precision, we instruct GPT-4 to focus on the qualified code, using the detailed specification only as reference. The resulting high-level descriptions succinctly summarize the code's functionality (what it does) and provide a concise overview of the implementation process (how it works). This annotation phase is the most critical and challenging as it demands that the model captures the code's high-level semantics, requiring a profound understanding of Verilog. In current benchmarks and practical applications, users typically prompt the model with high-level functional descriptions rather than detailed specifications. Otherwise, they would need to invest significant effort in writing exhaustive implementation details, making the process time-consuming and requiring extensive expertise. For detailed prompts used in this annotation process, please refer to Appendix B. And a detailed explanation of why we discard Verilog modules or blocks exceeding 2048 tokens can be found in Appendix C.

Given the industrial-grade quality of the proprietary code, we engage professional hardware engineers to maintain high annotation standards. Adhering to rigorous industry-level standards, these experts ensure precise and accurate annotations, capturing intricate details and significantly enhancing the dataset's value for advanced applications. Unlike GPT-generated annotations, these human-annotated ones incorporate an additional layer of granularity with medium-detail block descriptions. For detailed annotation standards and processes, please refer to Appendix D.

Table 1: The overall statistics of the annotation results for our dataset.

| Comment Level | Granularity | Count |
|---|---|---|
| Line Level | N/A | 434697 |
| Block Level | High-level Description | 892 |
| | Medium-Detail Description | 1306 |
| | Detailed Description | 894 |
| Module Level | High-level Functional Description | 59448 |
| | Detailed Specification | 59503 |

We present the overall statistics of the annotation results in Table 1. Additionally, Figure 2 illustrates an example of our comprehensive annotation for a complete Verilog module. Notably, the overall dataset encompassing descriptions of various details across multiple levels is used for training. A similar work to ours is the MG-Verilog dataset introduced by Zhang et al. (2024), including 11,000 Verilog code samples and corresponding natural language descriptions at various levels of details. However, it has several limitations compared to ours. Firstly, MG-Verilog is relatively small in size and lacks proprietary Verilog code, which diminishes its diversity and applicability. Secondly, it employs direct annotation rather than the CoT approach, which we have found to enhance annotation accuracy as demonstrated in Section 3.3. Besides, our annotation is more comprehensive than that of MG-Verilog, which lacks granularity. We cover line, block, and module levels with both detailed and high-level descriptions, ensuring a strong alignment between natural language and Verilog code. Lastly, MG-Verilog relies on the open-source LLM LLaMA2-70B-Chat for annotation, whereas we use the SOTA LLM GPT-4. In Section 4.4, we demonstrate that LLaMA2-70B-Chat has a poor understanding of Verilog code, leading to inferior annotation quality in MG-Verilog.

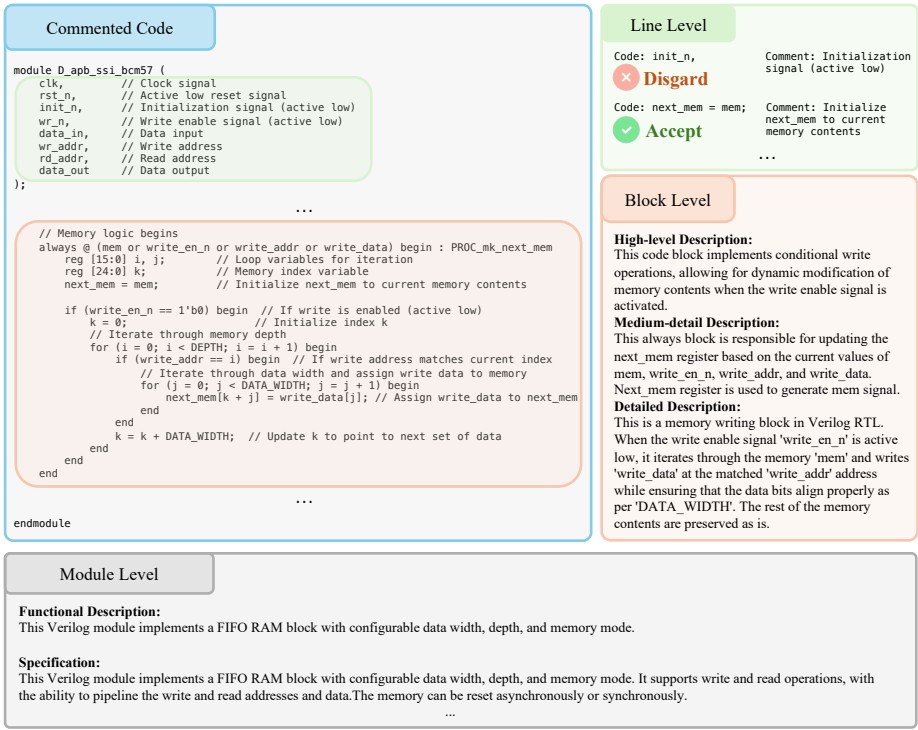

Figure 2: An example of our comprehensive annotation for a complete Verilog module.

## 3.3 DATASET EVALUATION

To ensure the quality of our dataset, we assess annotations generated from the CoT process. We randomly sample 200 Verilog modules and engage four professional Verilog designers to evaluate the accuracy of annotations at various levels. This human evaluation indicates that annotations describing high-level functions achieve an accuracy of $91\%$, while those providing detailed specifications attain an accuracy of $88\%$. For line-level annotations, the accuracy is $98\%$. Additionally, we compare the CoT method with the direct annotation approach, where annotations are generated straightforwardly from the original code. This direct annotation method yields only a $67\%$ accuracy, highlighting the significant advantage of integrating the CoT process.

Recent studies in natural language processing (NLP) have demonstrated that LLMs fine-tuned with synthetic instruction data can better understand natural language instructions and show improved alignment with corresponding tasks (Wang et al., 2022; Ouyang et al., 2022; Taori et al., 2023). It is important to note that in our work, we also utilize data generated by language models for fine-tuning, including annotations at various levels. While not all annotations are perfectly accurate, we achieve a commendable accuracy of approximately $90\%$. Motivated by Wang et al. (2022), we treat those inaccuracies as data noise, and the fine-tuned model on this dataset still derives significant benefits.

## 3.4 UNDERSTANDING BENCHMARK

As the first work to consider the task of Verilog understanding, we introduce a pioneering benchmark to evaluate LLMs' capabilities in interpreting Verilog code. This benchmark consists of 100 high-quality Verilog modules, selected to ensure comprehensive coverage of diverse hardware functionalities, providing a broad assessment scope across different types of hardware designs. We have engaged four experienced hardware engineers to provide precise annotations on each module's functionalities and the specific operations involved in their implementations. These initial annotations are then rigorously cross-verified by three additional engineers to guarantee accuracy and establish a high standard for future model evaluations. This benchmark fills a critical gap by providing a standardized means to assess LLMs on interpreting Verilog code and will be released later. For detailed examples included in the benchmark, please refer to Appendix E.

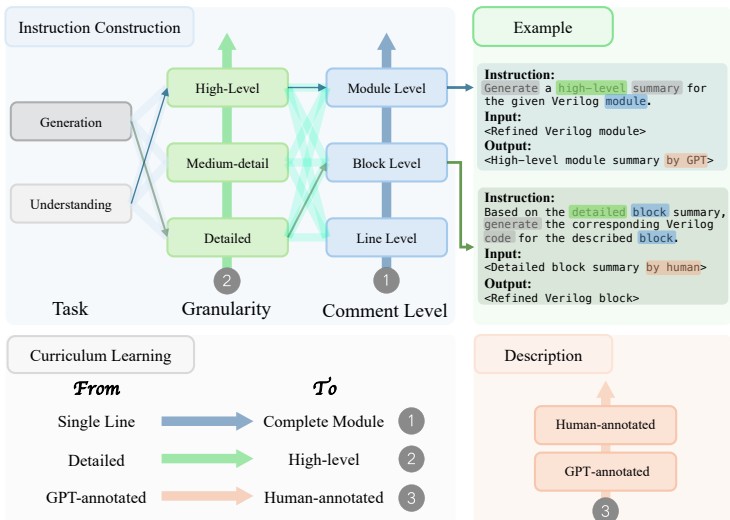

Figure 3: The overview of the instruction construction process and the curriculum learning strategy. For instruction construction, we integrate various settings, *e.g.*, task type, granularity, and comment level, to create tailored instructions for specific scenarios. The curriculum learning strategy involves three hierarchical stages: training progresses from line-level to module-level code (1st stage), transitioning from detailed to high-level descriptions at each level (2nd stage), and advancing from GPT-annotated to human-annotated descriptions for each granularity (3rd stage).

# 4  MODEL AND EVALUATION

In this section, we introduce DeepRTL and elaborate on the preparation of our instruction tuning dataset and how we adapt curriculum learning for training. Additionally, we detail the benchmarks and metrics used to evaluate our model's performance in both Verilog understanding and generation tasks. To accurately assess the semantic precision of the generated descriptions, we take the initiative to apply embedding similarity and GPT score for evaluation, which are designed to quantitatively measure the semantic similarity between the model's outputs and the ground truth.

## 4.1  MODEL

In our work, we have chosen to fine-tune CodeT5+ (Wang et al., 2023a), a family of encoder-decoder code foundation LLMs for a wide range of code understanding and generation tasks. CodeT5+ employs a "shallow encoder and deep decoder" architecture (Li et al., 2022), where both encoder and decoder are initialized from pre-trained checkpoints and connected by cross-attention layers. We choose to fine-tune CodeT5+ for its extensive pre-training on a vast corpus of software code, with the intent to transfer its acquired knowledge to hardware code tasks. Also, the model's flexible architecture allows for the customization of various training tasks, making it highly adaptable for specific downstream applications. Furthermore, CodeT5+ adopts an efficient fine-tuning strategy where the deep decoder is frozen and only the shallow encoder and cross-attention layers are allowed to train, significantly reducing the number of trainable parameters. Specifically, we have fine-tuned two versions of CodeT5+, codet5p-220m-bimodal[1] (CodeT5+-220m) and instructcodet5p-16b[2] (CodeT5+-16b), on our dataset, resulting in DeepRTL-220m and DeepRTL-16b, respectively. For more information on the model selection, please refer to Appendix F.

## 4.2  INSTRUCTION TUNING DATASET

During the fine-tuning process, we adopt the instruction tuning strategy to enhance the adaptability of LLMs, which is particularly effective when handling diverse types of data and tasks. Given that

---

[1] `https://huggingface.co/Salesforce/codet5p-220m-bimodal`
[2] `https://huggingface.co/Salesforce/instructcodet5p-16b`

our dataset features descriptions at multiple levels and our model is fine-tuned for both Verilog understanding and generation tasks, there is diversity in both the data types and tasks. To accommodate this diversity, we carefully design specific instructions for each scenario, ensuring the model can adjust its output to align with the intended instructions. Figure 3 illustrates how we combine various settings, *e.g.*, task type, granularity, and comment level, to construct tailored instructions for each specific scenario, fostering a structured approach to instruction-based tuning that optimizes the fine-tuning efficacy. For details on the instructions for different scenarios, please refer to Appendix G.

### 4.3 CURRICULUM LEARNING FOR DEEPRTL

We adapt curriculum learning for the fine-tuning process, leveraging our structured dataset that features descriptions of various details across multiple levels. Initially, the model is fine-tuned on line-level and block-level data, subsequently progressing to module-level data. At each level, we start by aligning the detailed specifications with the code before moving to the high-level functional descriptions. And fine-tuning typically starts with GPT-annotated data, followed by human-annotated data for each annotation granularity. Figure 3 provides an illustration of this process. We adopt such strategy because a particular focus is placed on aligning Verilog modules with their high-level functional descriptions, which poses the greatest challenge and offers substantial practical applications. This curriculum learning strategy enables the model to incrementally build knowledge from simpler to more complex scenarios. As a result, the models demonstrate impressive performance across both Verilog understanding and generation benchmarks. Note that we exclude the cases in the benchmarks from our training dataset. We primarily follow the instruction tuning script of CodeT5+[3] in the fine-tuning process, with a modification to expand the input context length to the maximum of 2048 tokens. We utilize the distributed framework, DeepSpeed, to efficiently fine-tune the model across a cluster equipped with eight NVIDIA A800 GPUs, each with 80GB of memory. During inference, we adjust the temperature to 0.8 for understanding tasks and to 0.5 for generation tasks, while other hyperparameters remain at their default settings to ensure optimal performance. Further details on the adopted curriculum learning strategy are provided in Appendix H.

### 4.4 UNDERSTANDING EVALUATION

For evaluating LLMs' capabilities in Verilog understanding, we utilize the benchmark introduced in Section 3.4. The evaluation measures the similarity between the generated descriptions and the ground truth summaries. Previous works usually use BLEU (Papineni et al., 2002) and ROUGE (Lin, 2004) scores for this purpose (Wang et al., 2023a). BLEU assesses how many $n$-grams, *i.e.*, sequences of $n$ words, in the machine-generated text appear in the reference text (focusing on precision). In contrast, ROUGE counts how many $n$-grams from the reference appear in the generated text (focusing on recall). However, both metrics primarily capture lexical rather than semantic similarity, which may not fully reflect the accuracy of the generated descriptions. To address this limitation, we take the initiative to apply embedding similarity and GPT score for evaluation. Embedding similarity calculates the cosine similarity between vector representations of generated and ground truth descriptions, using embeddings derived from OpenAI's text-embedding-3-large model. Meanwhile, GPT score uses GPT-4 to quantify the semantic coherence between descriptions by assigning a similarity score from 0 to 1, where 1 indicates perfect semantic alignment. These metrics provide a more nuanced evaluation by capturing the semantic essence of the descriptions, thus offering a more accurate assessment than previous methods. For details on the prompt used to calculate the GPT score, please refer to Appendix I.

### 4.5 GENERATION EVALUATION

To evaluate LLMs' capabilities in Verilog generation, we adopt the latest benchmark introduced by Chang et al. (2024a), which is an expansion based on the previous well-established RTLLM benchmark (Lu et al., 2024). The benchmark by Chang et al. (2024a) encompasses a broad spectrum of complexities across three categories: arithmetic, digital circuit logic, and advanced hardware designs. This benchmark extends beyond previous efforts by incorporating a wider range of more challenging and practical Verilog designs, thus providing a more thorough assessment of the models' capabilities in generating Verilog code.

---

[3]https://github.com/salesforce/CodeT5

Table 2: Evaluation results on Verilog understanding using the benchmark proposed in Section 3.4. BLEU-4 denotes the smoothed BLEU-4 score, and Emb. Sim. represents the embedding similarity metric. Best results are highlighted in bold.

| Model | BLEU-4 | ROUGE-1 | ROUGE-2 | ROUGE-L | Emb. Sim. | GPT Score |
|---|---|---|---|---|---|---|
| GPT-3.5 | 4.75 | 35.46 | 12.64 | 32.07 | 0.802 | 0.641 |
| GPT-4 | 5.36 | 34.31 | 11.31 | 30.66 | 0.824 | 0.683 |
| o1-preview | 6.06 | 34.27 | 12.25 | 31.01 | 0.806 | 0.643 |
| CodeT5+-220m | 0.28 | 7.10 | 0.34 | 6.18 | 0.313 | 0.032 |
| CodeT5+-16b | 0.10 | 1.37 | 0.00 | 1.37 | 0.228 | 0.014 |
| LLaMA2-70B-Chat | 2.86 | 28.15 | 10.09 | 26.12 | 0.633 | 0.500 |
| DeepRTL-220m-direct | 11.99 | 40.05 | 20.56 | 37.09 | 0.793 | 0.572 |
| DeepRTL-16b-direct | 11.06 | 38.12 | 18.15 | 34.85 | 0.778 | 0.533 |
| DeepRTL-220m | 18.66 | **47.69** | **29.49** | 44.02 | **0.837** | **0.705** |
| DeepRTL-16b | **18.94** | 47.27 | 29.46 | **44.13** | 0.830 | 0.694 |

The evaluation focuses on two critical aspects: syntax correctness and functional accuracy. We use the open-source simulator iverilog (Williams & Baxter, 2002) to assess both syntactic and functional correctness of Verilog code generated by LLMs. For the evaluation metric, we adopt the prevalent Pass@$k$ metric, which considers a problem solved if any of the $k$ generated code samples pass the compilation or functional tests (Pei et al., 2024). For this study, we set $k$ values of 1 and 5, where a higher Pass@$k$ score indicates better model performance. To further delineate the models' capabilities, we track the proportion of cases that pass out of 5 generated samples and compute the average as the success rate. For syntax correctness, this success rate measures the proportion of code samples that successfully compile and, for functional accuracy, the fraction that passes unit tests.

## 5 Experimental Results

### 5.1 Baseline Models

For the baseline models, we select OpenAI's GPT-4-turbo (GPT-4) and GPT-3.5-turbo (GPT-3.5), as well as the o1-preview model, OpenAI's latest reasoning model designed to address complex problems across diverse domains, including programming. These models are chosen for their status as the most advanced general-purpose LLMs currently available, with demonstrated excellence in Verilog generation (Chang et al., 2024b; Thakur et al., 2024; Liu et al., 2024). For a comparison with models specifically fine-tuned on Verilog, please refer to Appendix J.

### 5.2 Verilog Understanding

As shown in Table 2, DeepRTL consistently outperforms GPT-4 across all evaluation metrics. Traditional metrics like BLEU and ROUGE offer inconsistent assessments due to their inability to capture semantic similarity accurately: while DeepRTL-16b excels in BLEU-4 and ROUGE-L, DeepRTL-220m leads in ROUGE-1 and ROUGE-2. In contrast, embedding similarity and GPT score provide a more accurate assessment of the models' capabilities in understanding Verilog code. Compared to CodeT5+, the performance of DeepRTL-direct, which is trained directly without curriculum learning, highlights the effectiveness of our dataset. And the subsequent improvements when employing the curriculum learning strategy underscore its benefits. Additionally, the poor performance of LLaMA2-70B-Chat underscores the unreliability of the MG-Verilog annotations Zhang et al. (2024). To further validate our model's performance, we have conducted human evaluations, which show that DeepRTL-220m, GPT-4, and o1-preview achieve accuracies of 78%, 72%, and 67%, respectively. These results align closely with the embedding similarity and GPT score metrics, further affirming the effectiveness of these evaluation methods. We find that DeepRTL-220m outperforms DeepRTL-16b, likely due to the CodeT5+-220m's pre-training on a large corpus of paired software code and natural language data, which fosters better alignment between code and language. In contrast, CodeT5+-16b is primarily pre-trained on software code data and then fine-tuned on synthetic instruction data, lacking robust code-language alignment. Moreover, despite its smaller size, DeepRTL-220m surpasses many billion-parameter models, underscoring the high quality of our dataset and the effectiveness of the curriculum learning strategy.

Table 3: Evaluation results on Verilog generation. Each cell displays the percentage of code samples, out of five trials, that successfully pass compilation (syntax column) or functional unit tests (function column). Best results are highlighted in bold.

| Benchmark | | GPT-3.5 | | GPT-4 | | o1-preview | | DeepRTL-220m | | DeepRTL-16b | |
|---|---|---|---|---|---|---|---|---|---|---|---|
| | | syntax | function | syntax | function | syntax | function | syntax | function | syntax | function |
| Logic | Johnson_Counter | 40% | 0% | 100% | 0% | 100% | 0% | 100% | 0% | 100% | 0% |
| | alu | 0% | 0% | 0% | 0% | 0% | 0% | 0% | 0% | 0% | 0% |
| | edge_detect | 60% | 20% | 100% | 100% | 100% | 0% | 100% | 0% | 100% | 0% |
| | freq_div | 100% | 0% | 100% | 0% | 100% | 0% | 100% | 0% | 100% | 0% |
| | mux | 100% | 100% | 100% | 40% | 100% | 100% | 100% | 100% | 100% | 100% |
| | parallel2serial | 80% | 0% | 100% | 0% | 100% | 0% | 100% | 0% | 100% | 0% |
| | pulse_detect | 60% | 40% | 100% | 20% | 100% | 40% | 100% | 100% | 100% | 100% |
| | right_shifter | 60% | 60% | 100% | 100% | 100% | 100% | 100% | 100% | 100% | 100% |
| | serial2parallel | 60% | 0% | 100% | 0% | 100% | 20% | 100% | 0% | 100% | 0% |
| | width_8to16 | 100% | 0% | 20% | 0% | 100% | 0% | 100% | 0% | 100% | 0% |
| Arithmetic | accu | 100% | 0% | 40% | 0% | 100% | 0% | 100% | 0% | 100% | 0% |
| | adder_16bit | 40% | 0% | 20% | 20% | 40% | 40% | 100% | 0% | 60% | 0% |
| | adder_16bit_csa | 80% | 80% | 0% | 0% | 100% | 100% | 100% | 100% | 100% | 100% |
| | adder_32bit | 100% | 0% | 40% | 0% | 100% | 0% | 80% | 0% | 100% | 0% |
| | adder_64bit | 100% | 0% | 100% | 0% | 100% | 0% | 100% | 0% | 100% | 0% |
| | adder_8bit | 100% | 100% | 40% | 40% | 100% | 100% | 80% | 20% | 100% | 80% |
| | div_16bit | 0% | 0% | 0% | 0% | 0% | 0% | 0% | 0% | 0% | 0% |
| | multi_16bit | 80% | 0% | 100% | 20% | 100% | 100% | 100% | 0% | 100% | 0% |
| | multi_booth | 100% | 0% | 60% | 0% | 80% | 40% | 60% | 0% | 100% | 0% |
| | multi_pipe_4bit | 60% | 20% | 100% | 100% | 100% | 100% | 100% | 100% | 100% | 100% |
| | multi_pipe_8bit | 0% | 0% | 0% | 0% | 0% | 0% | 0% | 0% | 0% | 0% |
| Advanced | 1x2nocpe | 40% | 40% | 80% | 80% | 100% | 100% | 100% | 80% | 100% | 100% |
| | 1x4systolic | 100% | 100% | 100% | 100% | 100% | 100% | 100% | 100% | 100% | 100% |
| | 2x2systolic | 0% | 0% | 0% | 0% | 0% | 0% | 0% | 0% | 0% | 0% |
| | 4x4spatialacc | 0% | 0% | 0% | 0% | 0% | 0% | 0% | 0% | 0% | 0% |
| | fsm | 60% | 0% | 100% | 0% | 100% | 20% | 100% | 100% | 100% | 100% |
| | macpe | 0% | 0% | 0% | 0% | 0% | 0% | 0% | 0% | 0% | 0% |
| | 5state_fsm | 100% | 0% | 100% | 60% | 100% | 100% | 100% | 100% | 100% | 100% |
| | 3state_fsm | 20% | 0% | 80% | 20% | 100% | 100% | 100% | 100% | 100% | 100% |
| | 4state_fsm | 60% | 40% | 100% | 80% | 100% | 20% | 100% | 100% | 100% | 100% |
| | 2state_fsm | 80% | 20% | 100% | 80% | 100% | 0% | 100% | 20% | 0% | 0% |
| Success Rate | | 60.65% | 20.00% | 63.87% | 27.74% | **78.06%** | **38.06%** | **78.06%** | 36.13% | 76.13% | **38.06%** |
| Pass@1 | | 32.26% | 19.35% | 51.61% | 29.03% | **74.19%** | **35.48%** | 70.97% | 32.26% | **74.19%** | **35.48%** |
| Pass@5 | | **80.65%** | 35.48% | 77.42% | 45.16% | **80.65%** | **51.61%** | **80.65%** | 41.94% | 77.42% | 38.71% |

## 5.3 VERILOG GENERATION

Given the inferior performance of models like CodeT5+ and DeepRTL-direct in the Verilog understanding task, our comparison focuses on the GPT series models. As shown in Table 3, OpenAI's o1-preview, the latest model designed to tackle complex tasks including programming, achieves the highest performance across all metrics. Nevertheless, our DeepRTL model exhibits comparable performance to o1-preview on several metrics and significantly surpasses GPT-4 in syntax correctness, Pass@1 functional accuracy, and overall functional success rate. Notably, DeepRTL consistently generates highly accurate code in successful cases, often achieving a $100\%$ pass rate among the five generated samples, which underscores its reliability for practical applications. Furthermore, considering that OpenAI's models benefit from vast parameter sizes and extensive pre-training across diverse datasets, the performance of our more compact DeepRTL model is particularly impressive.

## 6 CONCLUSION

In this work, we introduce DeepRTL, a novel unified representation model that bridges Verilog understanding and generation. It is fine-tuned on a meticulously curated dataset featuring multi-level natural language descriptions of Verilog code, encompassing line, block, and module levels with both detailed and high-level functional descriptions. DeepRTL not only addresses the gaps in previous methods focused solely on Verilog code generation but also ensures strong alignment between Verilog code and natural language. Moreover, we establish the first benchmark for evaluating LLMs' capabilities in Verilog understanding. To overcome the limitations of traditional metrics like BLEU and ROUGE, which primarily assess lexical similarity, we apply embedding similarity and GPT score for evaluating the model's understanding capabilities. These metrics are designed to evaluate the semantic similarity of descriptions more accurately, thus better reflecting the precision of generated descriptions. By implementing a curriculum learning strategy, DeepRTL demonstrates superior performance in both Verilog understanding and generation tasks. Specifically, it surpasses the SOTA LLM, GPT-4, across all understanding metrics and achieves performance comparable to OpenAI's o1-preview in terms of syntax correctness and functional accuracy for Verilog generation.

ACKNOWLEDGMENTS

This work was supported in part by the General Research Fund of the Hong Kong Research Grants Council (RGC) under Grant No. 14212422 and 14202824, and in part by National Technology Innovation Center for EDA.

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

## A    INTRODUCTION OF VERILOG

Verilog is the most widely used hardware description language (HDL) for modeling digital integrated circuits. It enables designers to specify both the behavioral and structural aspects of hardware systems, such as processors, controllers, and digital logic circuits. Verilog operates at a relatively low level, focusing on gates, registers, and signal assignments—each representing physical hardware components. While Verilog supports behavioral constructs (*e.g.*, `if-else`, `case`) that are somewhat similar to software programming languages, their use is constrained by synthesizable coding styles required for hardware implementation. Verilog differs from software programming languages like Python and C++ in several key ways:

1. **Parallelism:** Verilog inherently models hardware's concurrnet nature, with multiple statements executing simultaneously. In contrast, software languages like Python typically follow a sequential execution model.

2. **Timing:** Timing is a fundamental concept in Verilog that directly influences how digital circuits are designed and simulated. Verilog relies on clocks to synchronize sequential logic behaviors, enabling the precise modeling of synthronous circuits. In contrast, software programming languages generally do not have an inherent need for explicit timing.

3. **Syntax and Constructs:** Verilog's syntax is tailored to describe the behavior and structure of digital circuits, reflecting the parallel nature of hardware. Key constructs of Verilog include:

   - **Modules:** The basic unit of Verilog, used to define a hardware block or component. Each module in Verilog encapsulates inputs, outputs, and internal logic, and modules can be instantiated within other modules, enabling hierarchical designs that mirror the complexity of real-world systems. And each module instantiation results in the generation of a corresponding circuit block.

   - **Always block:** In an `always` block, circuit designers can model circuits using high-level behavioral descriptions. However, this does not imply that a broad range of programming language syntax is available. In practice, Verilog supports only a limited subset of programming-like constructs, primarily `if-else` and `case` statements. Statements in multiple `always` blocks are executed in parallel and the resulting circuit continuously performs its operations.

   - **Sensitivity list:** In an `always` block, the sensitivity list specifies the signals that trigger the block's execution when they change.

   - **Assign statements:** `assign` statements are used to describe continuous assignments of signal values in parallel, reflecting the inherent concurrency of hardware.

   - **Registers (`reg`) and Wires (`wire`):** `reg` is used for variables that retain their values (*e.g.*, flip-flops or memory), and `wire` is used for connections that propagate values through the circuit.

   In contrast, software programming languages like C, Python, or Java employ a more conventional syntax for defining algorithms, control flow, and data manipulation. These languages use constructs like loops (`for`, `while`), conditionals (`if`, `else`), and functions or methods for structuring code, with data types such as integers, strings, and floats for variable storage.

## B    PROMPT DETAILS FOR CoT ANNOTATION

As shown in Figure 4, we present the detailed prompts used in our annotation process. For each task, we supplement the primary prompt with several human-reviewed input-output pair examples, serving as in-context learning examples to enhance GPT's understanding of task requirements and expectations. These examples will serve as guidance for the model to correctly interpret and execute tasks in accordance with the prompt, ensuring more accurate and contextually relevant outputs.

**System Prompt**

You are an expert in hardware design with a deep understanding of Verilog RTL codes. Now I want to train a large language model for code comment generation tasks specifically focused on Verilog RTL codes. And I want you to help me process the raw data, transforming it into suitable formats that can serve as inputs and outputs for the model.

**Qualified Code to Commented Code Prompt**

Add detailed comments to the following Verilog code, explaining each statement's purpose and functionality.

**Commented Code to Specification Prompt**

As an experienced digital IC design engineer, you've been tasked with creating a comprehensive design specification document for the RTL module provided above. Please analyze the RTL content and generate a detailed document with the following structure:
    1. Module Name: Clearly state the official name of the module.
    2. Module Functionality: Provide a thorough description of the module's primary functions and objectives. Include the types of signals it can process, its behavior under normal operating conditions, and any potential exception handling mechanisms.
    3. Input/Output Port Specification: List and describe all input and output signals, control signals, and status signals. For each, specify the data type, active level, timing relationships, purpose, and rules governing their changes.
    4. Internal Working Principles: Explain the module's internal mechanisms. Describe the logical implementation process, including:
        - Division of logic functionality (combinational and sequential logic)
        - Signal data flow and control flow
        - Special functional units (e.g., registers, arithmetic logic units, state machines)
    5. Algorithm and Logic Implementation Details: Provide a detailed explanation of the specific logic implemented in internal code blocks. This may involve algorithmic principles, implementation steps, and parameter configurations.
    6. Internal Signal Definitions: List and explain all signals defined within the module. Include their purposes, lifecycles, rules governing their changes, and relationships with other module components.
Please ensure that your explanation is clear, concise, and technically accurate, suitable for other engineers to understand and implement the design.

**Specification to Functional Description Prompt**

Please act as a professional Verilog designer. Your task is to summarize a concise yet accurate functional description of the provided Verilog code ('code' part), along with its specification ('specification' part). This description should offer a detailed explanation of the overall functionality, highlighting the key elements and operations of the code.
Summarize the functionality of the Verilog code using the 'code' part as the primary reference, with the 'specification' part serving as supplementary information. In case of any conflicts between the two, prioritize the information from the 'code' part.
When summarizing the description, keep the following points in mind:
    1. Focus on clarity and precision, ensuring that the description is both easy to understand and technically accurate.
    2. Use professional terminology related to Verilog/EDA/RTL.
    3. Avoid detailed operation comments, and instead focus on the overall functionality and purpose of the module.
    4. Only include information that is explicitly mentioned or implemented within the code. Do not add extraneous information or make assumptions about the module's characteristics or behavior that are not directly evident from the code.
    5. If submodules are found in the input code, provide a combined description of the general functionalities of all submodules. Avoid detailed internal logic descriptions within submodules and use qualifiers like "maybe" to prevent specific functional assumptions based on names. Integrate the functionality descriptions of all submodules into overall module functional description.
    6. Avoid speculation on specific product models, such as GPU manufacturer and model numbers.
    7. Do not mention any detailed internal logic or variable transformations.
The description should include answers to the following question with professional words/terms in Verilog/EDA/RTL:
Based on the functionality implemented by the provided Verilog code, what is the professional or functional term for this code? Please ensure the reply accurately reflects the specific operation or module described by the code.
If the code cannot be described in terms of functionality, such as in cases of interface definitions or simple variable definitions, do not infer based on variable or module names or specification I provided. Instead, format the output to end with "It does not perform any logical functions.", refer to the following example for generating the response:
Must respond in one complete sentence and do not use bullet points or numbered lists. Try to keep the length of answer under 100 tokens. Format the output to begin with: 'The module ... implements ...'.
Here's an example output format for your reference:
The module [module name] implements [overall function description].[Optional: If it cannot be described in terms of functionality, such as in cases of interface definitions or simple variable definitions] It does not perform any logical functions.

**Filtering Line-level Description Prompt**

You are an expert in hardware design with a deep understanding of Verilog RTL codes. Now I want to train a large language model for code comment generation tasks specifically focused on Verilog RTL codes.
Your task is to evaluate the relevance of the code and its corresponding comment. Please note that the comment should be directly related to the code, and the comment should be explicitly stated in the code I provide.
Do not include any speculative information, such as inferring functionality from variable names that the code itself does not explicitly show. If so, generate "*True*", otherwise, generate "*False*". Your answer should strictly be "*True*" or "*False*", no other content is allowed to be generated.

Figure 4: Detailed prompts used in the CoT annotation process.

## C   DISCARDING VERILOG CODE EXCEEDING 2048 TOKENS

In the main submission, we state that Verilog modules and blocks exceeding 2048 tokens are excluded, as 2048 is the maximum input length supported by CodeT5+. Beyond this limitation, several additional factors motivate this decision:

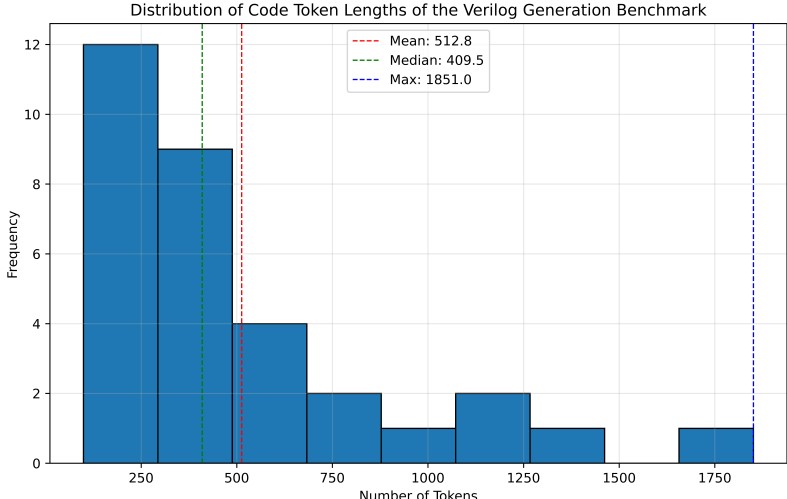

Figure 5: The distribution of the token lengths of the generation benchmark by Chang et al. (2024a).

1. **Generation Capabilities of Existing LLMs Are Limited to Small Designs**

   Existing benchmarks for Verilog generation, including the one used in our work (Chang et al., 2024a), do not include designs exceeding 2048 tokens, with the maximum token length observed in the benchmark being 1851. As shown in Table 3 of the main submission, even the state-of-the-art LLM, o1-preview, is capable of accurately generating only simple designs and struggles with more complex ones. Figure 5 illustrates the token length distribution across the benchmark, further justifying our decision to exclude Verilog modules and blocks exceeding 2048 tokens.

2. **Segmentation as a Common Practice**

   Segmenting longer code into smaller chunks that fit within the predefined context window and discarding those that exceed it is a widely accepted practice in both Verilog-related research (Chang et al., 2024b; Pei et al., 2024) and studies on software programming language (Wang et al., 2023a). This approach ensures compatibility with current LLMs while maintaining the integrity and usability of the dataset. It is worth noting that the default maximum sequence length in CodeT5+ is 512 tokens, and our work extends this limit to 2048 tokens to better accommodate Verilog designs.

3. **Empirical Findings and Practical Challenges**

   Our experiments reveal an important empirical observation: existing LLMs, such as GPT-4, consistently produce accurate descriptions for shorter Verilog modules but struggle with correctness when handling longer ones. Specifically, During the annotation process, we divide the dataset into two sections: Verilog designs with fewer than 2048 tokens, and designs with token lengths between 2048 and 4096 tokens. Our human evaluation finds that descriptions for Verilog designs with fewer than 2048 tokens are approximately 90% accurate, while descriptions for designs with token lengths between 2048 and 4096 tokens have accuracy rates of only 60%–70%. And accuracy further decreases for designs exceeding 4096 tokens. Since our datasets rely on LLM-generated annotations, restricting the dataset to Verilog modules within the 2048-token limit helps maintain the quality and accuracy of annotations. This, in turn, facilitates higher-quality dataset creation and more efficient fine-tuning. For the potential negative impact of incorporating Verilog designs larger than 2048 tokens, please refer to Appendix K. And we examine the impact of varying context window lengths in Appendix L.

## D  STANDARDS AND PROCESSES FOR MANUAL CODE ANNOTATION

Given the industrial-grade quality of the proprietary code, we employ professional hardware engineers for manual annotation. We have established the following standards and processes to guide engineers in crafting accurate and detailed descriptions with example annotations shown in Figure 6:

1. **Standards:** The hardware engineers are required to provide descriptions at both the module and block levels.

   - For module-level descriptions, two levels are defined:
     i. **H (High-level):** The role of this module in the overall design (IP/Chip).
     ii. **D (Detailed):** What functions this module performs (overview) and how it is implemented (implementation details). This description should adhere to a top-down structure and consist of approximately 2-5 sentences.

     **Note:** If the summary statements for H and D are identical, both must be provided.
   - For block-level descriptions, particularly `always` blocks, descriptions are required at three distinct levels:
     i. **H (High-level):** The role of this block in the overall design (*e.g.*, across modules).
     ii. **M (Medium-detail):** Contextual explanations.
     iii. **D (Detailed):** Descriptions specific to the block following a top-down structure. If details are absent, they may be omitted; do not guess based on signal names.

2. **Processes:** Initially, we provide engineers with a set of descriptions generated by GPT-4 for reference. They are then expected to revise and enhance these GPT-generated descriptions using their expertise and relevant supplementary materials, such as README files and register tables.

---

**D_apb_uart_bcm57.v**

*Module-level comment*

```
// H: This module implements the UART FIFO's read and write logic.
// D: This Verilog module implements a parameterized RAM with configurable data width, depth, and memory mode. It supports both read
and write operations, with the ability to pipeline the read and write signals based on the memory mode. The module includes
combinational processes for generating the next memory state and a read multiplexer function for selecting the read data. It also has
a sequential process for updating the memory and pipelining signals based on the clock and reset signals.

module D_apb_uart_bcm57 (
        clk,
        rst_n,
        init_n,
        wr_n,
        data_in,
        wr_addr,
        rd_addr,
        data_out
);
                                                    ...
```

*Block-level comment*

```
// H: This always block implements the logic of writing to the FIFO buffer when receiving a character or preparing for transmitting a
UART character.
// M: This always block is used to implement the logic of writing to the RAM, according to the data width and memory depth.
// D: This Verilog code is designed to manage the update process for a memory array named 'next_mem' with conditional write
operations. Initially, 'next_mem' is assigned the value of 'mem'. A write operation to 'next_mem' is only performed if 'write_en_n' is
active low. The memory address matching 'write_addr' is updated with the value of 'write_data'. Here, data is written bit-by-bit to
'next_mem' in a nested loop construct, maintaining the exact sequence as in 'write_data'. The address pointer 'k' increments by
'DATA_WIDTH' after each iteration.

always @(mem or write_en_n or write_addr or write_data) begin : PROC_mk_next_mem
  reg [15:0] i, j;
  reg [24:0] k;
  next_mem = mem;

  if (write_en_n == 1'b0) begin
    k = 0;
    for (i = 0; i < DEPTH; i = i + 1) begin
      if (write_addr == i) begin
        for (j = 0; j < DATA_WIDTH; j = j + 1) begin
          next_mem[k + j] = write_data[j];
        end
      end
      k = k + DATA_WIDTH;
    end
  end
end
                                                    ...
endmodule
```

Figure 6: Human-annotated examples for the proprietary code.

# E   EXAMPLES OF VERILOG UNDERSTANDING BENCHMARK

To construct a high-quality benchmark, we first remove comments from the original code, and then submit it to experienced hardware engineers for annotation, ultimately producing the code and description pairs as shown in Figure 7.

```
module beh_vlog_ff_ce_clr_v8_2 (
    Q,
    C,
    CE,
    CLR,
    D
);
parameter INIT = 0;
localparam FLOP_DELAY = 100;
output Q;
input C;
input CE;
input CLR;
input D;
reg Q;
initial Q = 1'b0;
always @(posedge C)
    if (CLR)
        Q <= 1'b0;
    else if (CE)
        Q <= #FLOP_DELAY D;
endmodule
```

```
module pcie_7x_v1_3_fast_cfg_init_cntr #(
    parameter PATTERN_WIDTH = 8,
    parameter INIT_PATTERN  = 8'hA5,
    parameter TCQ           = 1
)
(
    input                        clk,
    input                        rst,
    output reg [PATTERN_WIDTH-1:0] pattern_o
);

always @(posedge clk) begin
    if(rst) begin
        pattern_o <= #TCQ
{PATTERN_WIDTH{1'b0}};
    end else begin
        if(pattern_o != INIT_PATTERN) begin
            pattern_o <= #TCQ pattern_o + 1;
        end
    end
end
endmodule
```

```
module Event_Pulse(
    input in,
    input clk,
    output rising_edge,
    output falling_edge,
    output both_edges
);

    reg [1:0] reg_i = 2'b0;
    assign rising_edge = (~reg_i[1]) &
reg_i[0];
    assign falling_edge =  reg_i[1]
&(~reg_i[0]);
    assign both_edges = ((~reg_i[1]) &
reg_i[0]) | ( reg_i[1] & (~reg_i[0]));

    always @(posedge clk)
    begin
        reg_i[0] <= in;
        reg_i[1] <= reg_i[0];
    end
endmodule
```

**Qualified code**

The module
`beh_vlog_ff_ce_clr_v8_2`
implements a D-type flip-flop
with clock enable and
asynchronous clear
functionality, incorporating a
simulation delay for data
capture on the rising edge of
the clock."

The module
`pcie_7x_v1_3_fast_cfg_init_cntr`
implements a counter that operates on the
rising edge of the `clk` signal, resetting
`pattern_o` to zeros on `rst` assertion,
incrementing it by 1 on deassertion if not
equal to `INIT_PATTERN`, with a delay
specified by `TCQ`.

The module `Event_Pulse` implements a
detector for rising and falling edge
transitions of the input signal,
generating separate outputs for rising
edges, falling edges, and both edges
based on the sampled input signal at
the rising clock edge.

**Manually verified description**

Figure 7: Examples from the Verilog understanding benchmark.

## F  MODEL SELECTION

In this work, we choose CodeT5+, a family of encoder-decoder code foundation models, as the base model for training DeepRTL for two primary reasons. First, as we aim to develop a unified model for Verilog understanding and generation, T5-like models are particularly well-suited due to their ability to effectively handle both tasks, as evidenced by Wang et al. (2023a). Second, the encoder component of CodeT5+ enables the natural extraction of Verilog representations, which can be potentially utilized for various downstream tasks in EDA at the RTL stage. Examples include PPA (Power, Performance, Area) prediction, which estimates the power consumption, performance, and area of an RTL design, and verification, which ensures that the RTL design correctly implements its intended functionality and meets specification requirements. Both tasks are crucial in the hardware design process. This capability distinguishes it from decoder-only models, which are typically less suited for producing standalone, reusable intermediate representations. In future work, we plan to explore how DeepRTL can further enhance productivity in the hardware design process.

To further demonstrate the superiority of CodeT5+ as a base model, we fine-tune two additional models, deepseek-coder-1.3b-instruct[4] (deepseek-coder) (Guo et al., 2024) and Llama-3.2-1B-Instruct[5] (llama-3.2) (Dubey et al., 2024), using the same dataset as DeepRTL and the adopted curriculum learning strategy.

In Table 4 and Table 5, we present the performance of both the original base models and their fine-tuned counterparts on Verilog understanding and generation tasks. The improvement in performance from the original base models to the fine-tuned models highlights the effectiveness of our dataset and the curriculum learning-based fine-tuning strategy. Compared to the results in Table 2 and Table 3, the superior performance of DeepRTL-220m on both tasks, despite its smaller model size, underscores the architectural advantages of our approach.

## G  INSTRUCTIONS FOR DIFFERENT SCENARIOS

Figure 8 presents detailed instruction samples for different scenarios, following the instruction construction process illustrated in Figure 3. Additionally, it includes a special module-level task, which involves completing the source code based on the functional descriptions of varying granularity and the predefined module header.

---

[4]https://huggingface.co/deepseek-ai/deepseek-coder-1.3b-instruct
[5]https://huggingface.co/meta-llama/Llama-3.2-1B-Instruct

Table 4: Evaluation results on Verilog understanding using the benchmark proposed in Section 3.4. BLEU-4 denotes the smoothed BLEU-4 score, and Emb. Sim. represents the embedding similarity metric. Specifically, this table presents the performance of decoder-only models, where "long" indicates models fine-tuned on the dataset containing longer Verilog designs, and those fine-tuned specifically on Verilog. [†] indicates performance evaluated on designs shorter than $512$ tokens.

| Model | BLEU-4 | ROUGE-1 | ROUGE-2 | ROUGE-L | Emb. Sim. | GPT Score |
|---|---|---|---|---|---|---|
| deepseek-coder (original) | 1.04 | 21.43 | 4.38 | 19.77 | 0.678 | 0.557 |
| deepseek-coder (fine-tuned) | 11.96 | 40.49 | 19.77 | 36.14 | 0.826 | 0.664 |
| deepseek-coder (long) | 11.27 | 40.28 | 18.95 | 35.93 | 0.825 | 0.649 |
| llama-3.2 (original) | 0.88 | 19.26 | 3.60 | 17.64 | 0.615 | 0.449 |
| llama-3.2 (fine-tuned) | 12.11 | 39.95 | 19.47 | 35.29 | 0.825 | 0.620 |
| llama-3.2 (long) | 11.32 | 39.60 | 18.67 | 34.94 | 0.814 | 0.610 |
| RTLCoder | 1.08 | 21.83 | 4.68 | 20.30 | 0.687 | 0.561 |
| VeriGen | 0.09 | 6.54 | 0.35 | 6.08 | 0.505 | 0.311 |
| DeepRTL-220m-512[†] | 14.98 | 44.27 | 23.11 | 40.08 | 0.780 | 0.567 |
| DeepRTL-220m[†] | 18.74 | 48.41 | 29.82 | 45.01 | 0.855 | 0.743 |

Table 5: Evaluation results on Verilog generation. Each cell displays the percentage of code samples, out of five trials, that successfully pass compilation (syntax column) or functional unit tests (function column). This table presents the performance of decoder-only models, where "o" denotes the original model and "f" denotes the fine-tuned model.

| Benchmark | | deepseek-coder (o) | | deepseek-coder (f) | | llama-3.2 (o) | | llama-3.2 (f) | |
|---|---|---|---|---|---|---|---|---|---|
| | | syntax | function | syntax | function | syntax | function | syntax | function |
| Logic | Johnson_Counter | 100% | 0% | 100% | 0% | 100% | 0% | 100% | 0% |
| | alu | 0% | 0% | 0% | 0% | 0% | 0% | 0% | 0% |
| | edge_detect | 60% | 0% | 80% | 20% | 60% | 0% | 80% | 0% |
| | freq_div | 80% | 0% | 100% | 0% | 80% | 0% | 100% | 0% |
| | mux | 60% | 0% | 100% | 100% | 60% | 0% | 60% | 60% |
| | parallel2serial | 80% | 0% | 100% | 0% | 80% | 0% | 100% | 0% |
| | pulse_detect | 60% | 0% | 80% | 40% | 60% | 20% | 60% | 40% |
| | right_shifter | 20% | 0% | 80% | 80% | 20% | 0% | 40% | 40% |
| | serial2parallel | 100% | 0% | 100% | 0% | 100% | 0% | 100% | 0% |
| | width_8to16 | 100% | 0% | 100% | 0% | 100% | 0% | 100% | 0% |
| Arithmetic | accu | 80% | 0% | 100% | 0% | 80% | 0% | 100% | 0% |
| | adder_16bit | 20% | 0% | 40% | 20% | 20% | 0% | 20% | 20% |
| | adder_16bit_csa | 0% | 0% | 0% | 20% | 0% | 20% | 20% | 20% |
| | adder_32bit | 0% | 0% | 20% | 0% | 0% | 0% | 20% | 20% |
| | adder_64bit | 0% | 0% | 20% | 0% | 0% | 0% | 40% | 0% |
| | adder_8bit | 40% | 0% | 80% | 20% | 40% | 0% | 60% | 20% |
| | div_16bit | 0% | 0% | 20% | 0% | 0% | 0% | 0% | 0% |
| | multi_16bit | 60% | 0% | 80% | 0% | 60% | 0% | 80% | 0% |
| | multi_booth | 40% | 0% | 60% | 0% | 40% | 0% | 60% | 0% |
| | multi_pipe_4bit | 100% | 0% | 100% | 100% | 100% | 0% | 100% | 100% |
| | multi_pipe_8bit | 0% | 0% | 0% | 0% | 0% | 0% | 0% | 0% |
| Advanced | 1x2nocpe | 60% | 0% | 20% | 40% | 60% | 20% | 60% | 20% |
| | 1x4systolic | 20% | 0% | 100% | 100% | 20% | 0% | 20% | 20% |
| | 2x2systolic | 0% | 0% | 0% | 0% | 0% | 0% | 0% | 0% |
| | 4x4spatialacc | 0% | 0% | 0% | 0% | 0% | 0% | 0% | 0% |
| | fsm | 80% | 0% | 100% | 100% | 80% | 0% | 100% | 100% |
| | macpe | 0% | 0% | 0% | 0% | 0% | 0% | 0% | 0% |
| | 5state_fsm | 80% | 0% | 100% | 20% | 80% | 0% | 100% | 100% |
| | 3state_fsm | 0% | 0% | 100% | 80% | 20% | 20% | 100% | 100% |
| | 4state_fsm | 80% | 0% | 100% | 40% | 80% | 20% | 100% | 20% |
| | 2state_fsm | 60% | 0% | 100% | 20% | 60% | 0% | 100% | 20% |
| Success Rate | | 44.52% | 0.00% | 63.87% | 25.81% | 45.16% | 3.23% | 58.71% | 22.58% |
| Pass @ 1 | | 12.90% | 0.00% | 61.29% | 22.58% | 12.90% | 0.00% | 54.84% | 19.35% |
| Pass @ 5 | | 67.74% | 0.00% | 80.65% | 48.39% | 70.97% | 16.13% | 80.65% | 48.39% |

# H  FURTHER EXPLANATION OF THE ADOPTED CURRICULUM LEARNING STRATEGY

Our dataset includes three levels of annotation: line, block, and module, with each level containing descriptions that span various levels of detail—from detailed specifications to high-level functional

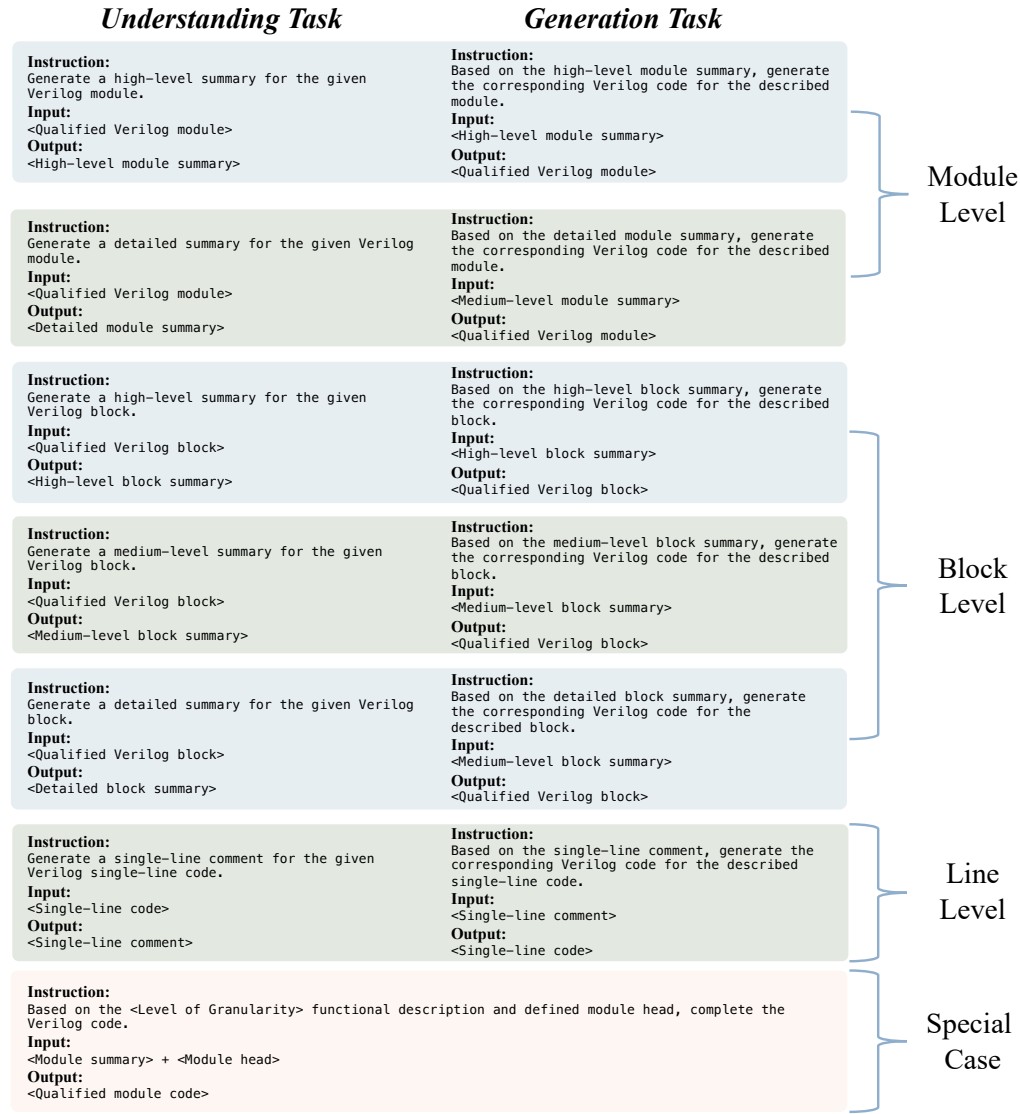

Figure 8: Instruction tuning data samples for different scenarios.

descriptions. And the entire dataset is utilized for training. To fully leverage the potential of this dataset, we employ a curriculum learning strategy, enabling the model to incrementally build knowledge by starting with simpler cases and advancing to more complex ones.

The curriculum learning strategy involves transitioning from more granular to less granular annotations across hierarchical levels, which can be conceptualized as a tree structure with the following components (as shown in Figure 9):

1. **Hierarchical Levels (First Layer)**

    The training process transitions sequentially across the three hierarchical levels—line, block, and module. Each level is fully trained before moving to the next, ensuring a solid foundation at simpler levels before addressing more complex tasks.

2. **Granularity of Descriptions (Second Layer)**

    Within each hierarchical level, the annotations transition from detailed descriptions to high-level descriptions. This progression ensures that the model learns finer details first and then builds an understanding of higher-level abstractions.

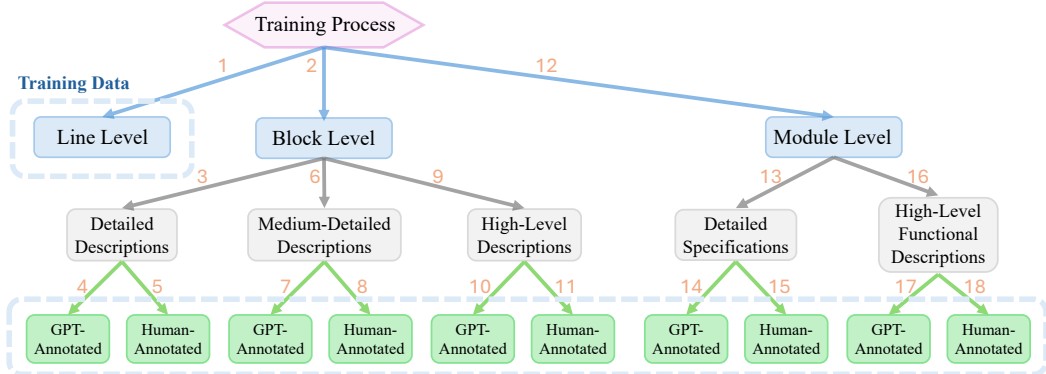

Figure 9: The adopted curriculum learning strategy visualized as a tree structure. Specifically, the terminals of the tree, enclosed by blue dotted boxes, represent specific training datasets. Our curriculum learning strategy follows a pre-order traversal of this tree structure.

3. **Annotation Source Transition (Third Layer)**

    At each level and granularity, training starts with GPT-annotated data and is followed by human-annotated data. This sequence leverages large-scale machine-generated annotations first and refines the model with high-quality, human-curated data.

4. **Instruction Blending**

    Each terminal node in this tree represents a specific training dataset, which blends tasks for Verilog understanding and Verilog generation. This enables the model to perform well across diverse tasks.

The training process mirrors a pre-order traversal of this tree structure:

1. Starting at the root, training begins with the line level.

2. The model progresses through the second layer (detailed, medium-detail, and high-level descriptions).

3. Within each granularity, training transitions through the third layer (GPT-annotated data first, followed by human-annotated data).

4. Once the line level is complete, the process repeats for the block level and then the module level.

To validate the effectiveness of this strategy, we conduct an ablation study where the model is trained on the entire dataset all at once without progression. The results, presented in Table 2 of the main submission, demonstrate that the curriculum learning strategy significantly outperforms this baseline approach. Moreover, to the best of our knowledge, this is one of the first applications of a curriculum-like training strategy in the code-learning domain. Unlike existing Verilog-related models that establish simple and weak alignments between natural language and Verilog code (Chang et al., 2024b), or general software code datasets like CodeSearchNet[6] (Husain et al., 2019) that only provide single-level docstring annotations, our approach incorporates multi-level and multi-granularity annotations in a structured training process.

## I    PROMPT FOR CALCULATING GPT SCORE

To calculate the GPT score, we input the model's generated descriptions (model_output) and the ground truth annotations (ground_truth) to GPT-4, using the prompt displayed in Figure 10. This metric is designed to assess the semantic accuracy of the generated functional descriptions.

---

[6]https://huggingface.co/datasets/code-search-net/code_search_net

You are a professional Verilog designer that needs to evaluate the similarity between two textual functional summaries describing Verilog code.
The first summary is the ground truth description of the Verilog code, and the second summary is the generated description of the Verilog code.
Please read the following summaries and provide a similarity score between 0 and 1 based on how similar the two summaries are in terms of describing the functionality of the Verilog code, where 0 means completely dissimilar and 1 means identical.
Note that you should strictly only output the score without any additional information.
Summary 1: {*ground_truth*}
Summary 2: {*model_output*}

Figure 10: Prompt used to calculate the GPT score.

## J    COMPARISON WITH MODELS SPECIFICALLY TRAINED ON VERILOG

To further demonstrate the superiority of DeepRTL, we conduct experiments comparing it with models specifically trained on Verilog. We do not select (Chang et al., 2024b; Zhang et al., 2024) for comparison, as their models are not open-sourced, and it is non-trivial to reproduce their experiments. Additionally, the reported performance in their original papers is either comparable to, and in some cases inferior to, that of GPT-3.5. In Table 4 and Table 6, we show the performance of two state-of-the-art Verilog generation models, RTLCoder-Deepseek-v1.1[7] (RTLCoder) (Liu et al., 2024) and fine-tuned-codegen-16B-Verilog[8] (VeriGen) (Thakur et al., 2024) on both Verilog understanding and generation benchmarks. It is noteworthy that RTLCoder is fine-tuned on DeepSeek-coder-6.7B, and VeriGen is fine-tuned on CodeGen-multi-16B, both of which have significantly larger parameter sizes than DeepRTL-220m. Despite this, the superior performance of DeepRTL-220m further underscores the effectiveness of our proposed dataset and the adopted curriculum learning strategy.

Table 6: Evaluation results on Verilog generation. Each cell displays the percentage of code samples, out of five trials, that successfully pass compilation (syntax column) or functional unit tests (function column). This table presents the performance of models specifically trained on Verilog.

| Benchmark | | RTLCoder | | VeriGen | |
|---|---|---|---|---|---|
| | | syntax | function | syntax | function |
| Logic | Johnson_Counter | 40% | 0% | 100% | 0% |
| | alu | 0% | 0% | 0% | 0% |
| | edge_detect | 100% | 100% | 100% | 20% |
| | freq_div | 60% | 0% | 100% | 0% |
| | mux | 60% | 40% | 80% | 20% |
| | parallel2serial | 60% | 0% | 100% | 0% |
| | pulse_detect | 20% | 0% | 40% | 0% |
| | right_shifter | 80% | 80% | 100% | 100% |
| | serial2parallel | 60% | 0% | 80% | 0% |
| | width_8to16 | 60% | 0% | 100% | 0% |
| Arithmetic | accu | 0% | 0% | 0% | 0% |
| | adder_16bit | 40% | 20% | 20% | 0% |
| | adder_16bit_csa | 80% | 80% | 0% | 0% |
| | adder_32bit | 80% | 0% | 0% | 0% |
| | adder_64bit | 40% | 0% | 40% | 0% |
| | adder_8bit | 80% | 40% | 40% | 40% |
| | div_16bit | 0% | 0% | 0% | 0% |
| | multi_16bit | 80% | 0% | 80% | 0% |
| | multi_booth | 20% | 0% | 20% | 0% |
| | multi_pipe_4bit | 60% | 20% | 80% | 20% |
| | multi_pipe_8bit | 0% | 0% | 0% | 0% |
| Advanced | 1x2nocpe | 40% | 40% | 100% | 100% |
| | 1x4systolic | 100% | 100% | 20% | 20% |
| | 2x2systolic | 0% | 0% | 0% | 0% |
| | 4x4spatialacc | 0% | 0% | 0% | 0% |
| | fsm | 100% | 60% | 80% | 20% |
| | macpe | 0% | 0% | 0% | 0% |
| | 5state_fsm | 60% | 40% | 80% | 0% |
| | 3state_fsm | 80% | 0% | 80% | 20% |
| | 4state_fsm | 80% | 0% | 80% | 20% |
| | 2state_fsm | 20% | 0% | 60% | 0% |
| Success Rate | | 48.39% | 20.00% | 50.97% | 12.26% |
| Pass @ 1 | | 41.94% | 16.13% | 48.39% | 9.68% |
| Pass @ 5 | | 77.42% | 35.48% | 70.97% | 32.26% |

---

[7]https://huggingface.co/ishorn5/RTLCoder-Deepseek-v1.1
[8]https://huggingface.co/shailja/fine-tuned-codegen-16B-Verilog

Table 7: Evaluation results on Verilog generation. Each cell displays the percentage of code samples, out of five trials, that successfully pass compilation (syntax column) or functional unit tests (function column). This table presents the performance of decoder-only models fine-tuned on the dataset containing longer Verilog designs.

| Benchmark | | deepseek-coder | | llama-3.2 | |
|---|---|---|---|---|---|
| | | syntax | function | syntax | function |
| Logic | Johnson_Counter | 100% | 0% | 100% | 0% |
| | alu | 0% | 0% | 0% | 0% |
| | edge_detect | 80% | 0% | 80% | 0% |
| | freq_div | 100% | 0% | 100% | 0% |
| | mux | 100% | 100% | 60% | 60% |
| | parallel2serial | 100% | 0% | 100% | 0% |
| | pulse_detect | 80% | 40% | 60% | 40% |
| | right_shifter | 80% | 80% | 40% | 40% |
| | serial2parallel | 100% | 0% | 100% | 0% |
| | width_8to16 | 100% | 0% | 100% | 0% |
| Arithmetic | accu | 100% | 0% | 100% | 0% |
| | adder_16bit | 20% | 20% | 20% | 20% |
| | adder_16bit_csa | 20% | 20% | 20% | 20% |
| | adder_32bit | 0% | 0% | 20% | 20% |
| | adder_64bit | 0% | 0% | 0% | 0% |
| | adder_8bit | 80% | 20% | 60% | 20% |
| | div_16bit | 20% | 0% | 0% | 0% |
| | multi_16bit | 80% | 0% | 80% | 0% |
| | multi_booth | 60% | 0% | 60% | 0% |
| | multi_pipe_4bit | 100% | 100% | 100% | 100% |
| | multi_pipe_8bit | 0% | 0% | 0% | 0% |
| Advanced | 1x2nocpe | 40% | 40% | 60% | 20% |
| | 1x4systolic | 20% | 20% | 20% | 20% |
| | 2x2systolic | 0% | 0% | 0% | 0% |
| | 4x4spatialacc | 0% | 0% | 0% | 0% |
| | fsm | 100% | 100% | 100% | 100% |
| | macpe | 0% | 0% | 0% | 0% |
| | 5state_fsm | 100% | 0% | 100% | 100% |
| | 3state_fsm | 80% | 80% | 100% | 100% |
| | 4state_fsm | 100% | 0% | 100% | 0% |
| | 2state_fsm | 100% | 20% | 100% | 20% |
| Success Rate | | 60.00% | 20.65% | 57.42% | 21.94% |
| Pass @ 1 | | 38.71% | 19.35% | 38.71% | 19.35% |
| Pass @ 5 | | 77.42% | 38.71% | 77.42% | 45.16% |

# K    NEGATIVE IMPACT OF INCORPORATING VERILOG DESIGNS EXCEEDING 2048 TOKENS

Notably, the maximum input length for DeepSeek-coder is 16k tokens, while for LLaMA-3.2, it is 128k tokens. To assess the potential negative impact of including Verilog designs exceeding 2048 tokens, we conduct an ablation study in which we do not exclude such modules for these two models and instead use the dataset containing longer designs for training. As shown in Table 7, and by comparing the results in Table 4, the performance of the fine-tuned models on both Verilog understanding and generation tasks significantly degrades compared to the results in Table 5, where these models are fine-tuned using the same dataset as DeepRTL. This further validates the rationale behind our decision to exclude Verilog modules and blocks exceeding 2048 tokens.

# L    ADDITIONAL EXPERIMENTS INVESTIGATING THE IMPACT OF VARYING CONTEXT WINDOW LENGTHS

To address concerns regarding the potential bias introduced by excluding examples longer than 2048 tokens, we investigate the impact of context window length. Specifically, we exclude all Verilog modules exceeding 512 tokens and use the truncated dataset to train a new model, DeepRTL-220m-512 utilizing the curriculum learning strategy, which has a maximum input length of 512 tokens. We then evaluate both DeepRTL-220m-512 and DeepRTL-220m on Verilog understanding benchmark samples, where the module lengths are below 512 tokens, and present the results in Table 4. For the generation task, DeepRTL-220m-512 shows near-zero performance, with nearly 0% accuracy for both syntax and functional correctness. This result refutes the concern that "a model accommodating longer context windows could potentially offer superior performance on the general task, but not for this tailored dataset," as it does not hold true in our case.

