# OpenReview forum: "DeepRTL: Bridging Verilog Understanding and Generation with a Unified Representation Model"
_ICLR.cc/2025/Conference — ICLR 2025 Spotlight_

### Official Review · Reviewer_vHqv · 2024-10-30

**Soundness:** 3
**Presentation:** 3
**Contribution:** 3
**Rating:** 8
**Confidence:** 5

**Summary:**

This paper introduces a novel dataset and model training for Verilog understanding and generation, as well as a new high-quality benchmark for the understanding task.

The authors provide a large Verilog dataset based on a large quantity of crawled open source code that is processed into code and natural language descriptions via GPT4, as well as a smaller amount of hand-curated code-description items from proprietary sources.\
They also introduce a new benchmark for Verilog understanding, consisting of 100 manually verified, high-quality code-description pairs.

For experiments, the authors train CodeT5+ -based models of sizes 220M and 16B on their newly introduced dataset, using "progressive training" and evaluate model performance in terms of Verilog understanding and generation capabilities.\
Experiments show that models trained in this manner outperform strong baselines on various metrics.

**Strengths:**

A novel dataset and benchmark for highly specialised programming code (Verilog); this might be interesting as it provides a new and interesting resource for a programming language that does not have as much attention as others such as Python, Jave, or C++.

**Weaknesses:**

Beyond the curation of an interesting new dataset, there is very limited novelty to this work; it seems like authors might be somewhat unfamiliar with the current state of the field of LLMs/Machine Learning, including ML for code:
- Fine-tuning a CodeT5 model on domain-specific code has been done.
- The "progressive training" is just curriculum learning, which is well-established in the field.
- Similarity scores based on vector similarity are as old as Word2Vec, if not older.
- Similarities/evaluations with LMs or LLMs (here "GPT Score") are well-established, e.g., see "LLM as a judge", BERT Score, etc.

This seems like it would be a very nice paper for a specialised Verilog/hardware spec conference, but may be of limited value for a venue like ICLR.

**Questions:**

- Why throw away dataset items that are longer than 2,048 tokens? It is true that this is the maximum input length for CodeT5+; however, why make a choice about the dataset based on the (essentially arbitrary) choice of model used in the specific experiments here?\
Modern LLMs, including open source ones such as Llama, have context sizes way beyond 2,048 tokens.

**Comment after Rebuttal**
I have adjusted my score to "marginally below acceptance threshold", rather than an outright "reject" based very good rebuttals.\
A agree with the authors on certain parts of my original criticism; however, the current manuscript requires some significant amount of re-work to be "acceptable", especially wrt. the original claims and presented support for decisions such as base model choice, and length restrictions of the dataset.

**Comment after further Rebuttal**
The authors have done an exceptional job at responding to reviewers' concerns, and addressed them in an updated manuscript. After the changes, I believe this paper is in a state that is acceptable; I have updated my scores accordingly.

---

> ### Author Response · Authors · 2024-11-25
> **Response to Reviewer vHqv (Part 1/4)**
>
> ### Q1: “Beyond the curation of an interesting new dataset, there is very limited novelty to this work.”
>
> R1: Thank you for your thoughtful feedback, and we appreciate that you think our work is a nice paper. Below, we provide a detailed explanation to highlight the unique challenges in building a foundation model for Verilog, clarify the key contributions of our work, and address the perceived limitations.
>
> **Challenges in Building a Foundation Model for Verilog**
>
> Building a foundation model for Verilog presents unique challenges due to the unique characteristics of the language and the scarcity of high-quality resources:
>
> - **Significant Differences from Software Programming Languages:**
>
>     Verilog is a hardware description language with constructs and semantics tailored specifically to hardware design. Unlike software programming languages, Verilog involves specialized paradigms such as concurrency, timing control, and hardware-specific constraints, making it nontrivial to directly transfer knowledge from existing software foundation models to Verilog.
>
> - **Data Scarcity:**
>
>     Verilog is a low-resource language underrepresented in conventional code datasets. As shown in [1], Verilog repositories contain orders of magnitude fewer files than those for general-purpose programming languages like Python or Java. This lack of data makes it challenging to gather the large-scale datasets typically required to train robust foundation models.
>
> - **Poor Dataset Quality:**
>
>     Existing Verilog datasets often suffer from weak alignment between natural language descriptions and Verilog code. This misalignment hinders a model's ability to learn accurate mappings between textual specifications and hardware designs, which is critical for Verilog understanding and generation. Without rich, well-annotated datasets, the potential of foundation models remains limited.
>
>
> **Main Contributions of Our Work**
>
> 1. **A High-Quality, Comprehensive Dataset:**
>
>     We introduce the first high-quality dataset that aligns Verilog code with rich, multi-level natural language descriptions. This comprehensive resource addresses the scarcity of well-annotated Verilog datasets, enabling both understanding and generation tasks for this specialized hardware description language.
>
> 2. **Meticulous Annotation Strategy**:
>
>     Recognizing the critical impact of dataset quality on model performance, we design a meticulous annotation strategy leveraging Chain-of-Thought (CoT). This ensures strong alignment between Verilog code and multi-level natural language descriptions, setting a new standard for datasets in the code-learning domain.
>
> 3. **A Unified Model for Verilog Understanding and Generation:**
>
>     We propose DeepRTL, the first unified model capable of both Verilog understanding and generation. Importantly, we are the first to consider the task of Verilog understanding, which is a critical task overlooked by previous works. In addition, we introduce the first benchmark specifically tailored to Verilog understanding.
>
> 4. **Progressive Training Strategy**:
>
>     Our progressive training strategy aligns with the principles of **curriculum learning**, introducing simpler concepts first and incrementally transferring knowledge to handle more complex scenarios.
>
>     To validate the effectiveness of this strategy, we conducted an ablation study where the model was trained on the entire dataset all at once without progression. As shown in Table 2 of the original manuscript, the progressive training strategy significantly outperforms this baseline. To the best of our knowledge, this is the first application of a curriculum-like training strategy in the code-learning domain.
>
>     Unlike existing Verilog models, which typically establish weak alignments between Verilog code and natural language annotations [1], or general software datasets like CodeSearchNet [2], which only provide single-level docstring annotations, our progressive strategy incorporates multi-level and multi-granularity annotations in a structured training process. This approach enables DeepRTL to achieve strong performance even with a lightweight 220M parameter model.
>
> 5. **Novel Evaluation Metrics**:
>
>     We introduce two new evaluation metrics, embedding similarity and GPT score, for assessing code understanding. These metrics capture semantic similarities between generated and ground truth descriptions more effectively than traditional metrics like BLEU and ROUGE. To the best of our knowledge, this is the first application of these metrics to evaluate the code understanding capabilities of LLMs, providing a more robust and reliable assessment framework for code-learning tasks.
>
> [1] Data is all you need: Finetuning LLMs for Chip Design via an Automated design-data augmentation framework. DAC 2024.
>
> [2] https://huggingface.co/datasets/code-search-net/code_search_net

---

> ### Author Response · Authors · 2024-11-25
> **Response to Reviewer vHqv (Part 2/4)**
>
> ### **Responses to Specific Weaknesses Raised by the Reviewer**
>
> 1. **"Fine-tuning a CodeT5 model on domain-specific code has been done."**
>
>     It is indeed common practice to adapt pre-trained models to domain-specific tasks. However, our work goes beyond simple fine-tuning by creating a high-quality dataset tailored to Verilog and demonstrating a unified model that bridges understanding and generation tasks. This has not been achieved before for Verilog, a specialized and under-resourced language.
>
> 2. **"The 'progressive training' is just curriculum learning, which is well-established in the field."**
>
>     While progressive training aligns with curriculum learning, this is, to the best of our knowledge, the first time it has been applied to the code learning domain. Our approach combines multi-level, multi-granularity annotations with structured training to handle the challenges posed by limited datasets and the unique nature of Verilog. The ablation study presented in Table 2 of the original manuscript highlights the significant gains achieved through this strategy, demonstrating its value in the code-learning domain.
>
> 3. **"Similarity scores based on vector similarity are as old as Word2Vec, if not older."**
>
>     While vector similarity methods have been used in NLP, their application to code-learning, specifically Verilog, is novel. Embedding similarity provides a robust way to evaluate semantic alignment between generated descriptions and ground truth summaries, addressing the limitations of traditional metrics like BLEU and ROUGE.
>
> 4. **"Similarities/evaluations with LMs or LLMs (here 'GPT Score') are well-established..."**
>
>     Although GPT-based evaluation frameworks like "LLMs as judges" or BERTScore are established in NLP, this is the first adaptation of such metrics for Verilog. Our work demonstrates their utility in evaluating the code understanding capabilities of LLMs for specialized domains like Verilog, filling an important gap in evaluation methods for code-learning tasks.
>
> 5. **"This seems like it would be a very nice paper for a specialized Verilog/hardware spec conference, but may be of limited value for a venue like ICLR."**
>
>     We respectfully disagree with this point. Our work not only addresses a critical gap in Verilog-related resources but also demonstrates broader implications for the machine learning community. Specifically:
>
>     - We establish a proof of concept for designing unified models tailored to under-resourced languages, showcasing how high-quality datasets and innovative training strategies can compensate for model size.
>     - We introduce new evaluation metrics and benchmarks that capture semantic understanding more effectively than traditional methods to code-learning tasks, inspiring further exploration in other specialized domains.
>
> Furthermore, works similar to ours—proposing novel datasets or fine-tuning LLMs for specific domains—have been successfully published in leading machine learning conferences [3][4]. This precedent highlights the relevance of our contributions to the broader ML community. We believe our contributions can inspire both the machine learning and electronic design automation communities to advance this field.
>
> We hope these clarifications highlight the key contributions and novelty of our work. We will revise the manuscript to make these points more explicit and welcome further discussions or suggestions from the reviewer.
>
> [3] BetterV: Controlled Verilog Generation with Discriminative Guidance, ICML 2024.
>
> [4] Mol-Instructions: A Large-Scale Biomolecular Instruction Dataset for Large Language Models. ICLR 2024.

---

> ### Author Response · Authors · 2024-11-25
> **Response to Reviewer vHqv (Part 3/4)**
>
> ### Q2: “Why throw away dataset items that are longer than 2,048 tokens? It is true that this is the maximum input length for CodeT5+; however, why make a choice about the dataset based on the (essentially arbitrary) choice of model used in the specific experiments here? Modern LLMs, including open source ones such as Llama, have context sizes way beyond 2,048 tokens.”
>
> R2: We thank the reviewer for this thoughful and valueable feedback. While it is true, as the reviewer points out, that 2,048 tokens represent the maximum input length for CodeT5+, our decision to exclude Verilog modules exceeding this threshold is motivated by additional factors:
>
> 1. **Generation Capabilities of Existing LLMs Are Limited to Small Designs**
>
>     The existing benchmarks for Verilog generation, including the one used in our work [5], do not include designs exceeding 2,048 tokens. The maximum token length observed in the benchmark is 1,851. As shown in Table 3 of the original manuscript, even the state-of-the-art LLM, o1-preview, is limited to generating simple designs accurately and lacks the capability to handle more complex designs. To further clarify why we exclude Verilog modules exceeding 2,048 tokens, we will include a figure in the revised manuscript that illustrates the token length distribution across the benchmark.
>
>     We also recognize the importance of evaluating models on Verilog code that exceeds the 2,048-token threshold, as real-world Verilog designs often surpass this limit. However, creating a benchmark tailored to longer examples presents significant challenges, particularly due to the lack of automated tools for generating testbenches for these extended cases.
>
> 2. **Segmentation as a Common Practice**
>
>     Segmenting longer code into smaller chunks that fit within the predefined context window and discarding those that exceed it is a widely accepted practice in both Verilog-related research ([1] and [3]) and studies on software programming languages [6]. This approach ensures compatibility with current LLMs while maintaining the integrity and usability of the dataset. It is worth noting that the default maximum sequence length in CodeT5+ is 512 tokens, and our work extends this limit to 2,048 tokens to better accommodate Verilog designs.
>
> 3. **Empirical Findings and Practical Challenges**
>
>     Our experiments reveal a key empirical observation: existing LLMs, such as GPT-4, consistently produce accurate descriptions for shorter Verilog modules but struggle with correctness when handling longer ones. Since our datasets rely on LLM-generated annotations, restricting the dataset to Verilog modules within the 2,048-token limit helps maintain the quality and accuracy of annotations. This ensures higher-quality dataset creation and facilitates efficient fine-tuning.
>
>
> However, we agree that developing and evaluating models capable of processing longer Verilog files is an essntial task as many real-world Verilog designs exceed this length. In future work, we plan to explore models with extended context lengths and evaluate their performance on datasets containing longer Verilog modules.
>
> **Choice of Base Model**
>
> The selection of CodeT5+ as the base model for DeepRTL is not made arbitrarily. Instead, we choose CodeT5+, a family of encoder-decoder code foundation models, for two primary reasons. First, as we aim to develop a unified model capable of both Verilog understanding and generation, T5-like models are particularly well-suited for this purpose, as evidenced by their ability to handle both tasks effectively [1].  Second, the encoder component of CodeT5+ enables the natural extraction of Verilog representations, which can be potentially utilized for various downstream tasks in Electronic Design Automation (EDA) at the RTL stage. Examples include PPA (Power, Performance, Area) prediction, which estimates the power consumption, performance, and area of an RTL design, and verification, which ensures that the RTL design adheres to its intended functionality and meets specification requirements. These are two critical tasks in the hardware design process. This capability distinguishes it from decoder-only models, which are typically less suited for producing standalone, reusable intermediate representations. In future work, we aim to further enhance DeepRTL’s productivity in the hardware design process by expanding its capabilities and evaluating its impact across additional EDA tasks.
>
> [1] Data is all you need: Finetuning LLMs for Chip Design via an Automated design-data augmentation framework. DAC 2024.
>
> [3] BetterV: Controlled Verilog Generation with Discriminative Guidance, ICML 2024.
>
> [5] Natural language is not enough: Benchmarking multi-modal generative AI for Verilog generation. ICCAD 2024.
>
> [6] CodeT5+: Open Code Large Language Models for Code Understanding and Generation. EMNLP 2023.

---

> ### Author Response · Authors · 2024-11-25
> **Response to Reviewer vHqv (Part 4/4)**
>
> **Comparison with Other Base Models with Different Architectures and Context Lengths**
>
> To further demonstrate the superiority of CodeT5+ as a base model, we fine-tune two additional models, deepseek-coder-1.3b-instruct [7] and Llama-3.2-1B-Instruct [8], using our proposed dataset and progressive training strategy. Notably, the maximum input length for deepseek-coder-1.3b-instruct is 16k tokens, and for Llama-3.2-1B-Instruct, it is 128k tokens. As a result, we did not exclude Verilog modules exceeding 2,048 tokens in these two cases.
>
> In the following tables, we present the performance of both the original base models and their fine-tuned counterparts on Verilog understanding and generation tasks, alongside the results from our DeepRTL-220m model. The improvement in performance from the original base models to the fine-tuned models highlights the effectiveness of our dataset and progressive fine-tuning strategy. Meanwhile, the superior performance of DeepRTL-220m on both tasks, despite its smaller model size, underscores the architectural advantages of our approach.
>
> We hope these experimental results can provide more insights into the impact of token length limitations and model architecture on final performance. These experimental results will be incorporated into the revised manuscript.
>
> | Understanding | BLEU-4 | ROUGE-1 | ROUGE-2 | ROUGE-L | Emb. Sim. | GPT Score |
> | --- | --- | --- | --- | --- | --- | --- |
> | deepseek-coder-1.3b-instruct (original) | 1.04 | 21.43 | 4.38 | 19.77 | 0.678 | 0.557 |
> | deepseek-coder-1.3b-instruct (fine-tuned) | 11.27 | 40.28 | 18.95 | 35.93 | 0.825 | 0.649 |
> | Llama-3.2-1B-Instruct (original) | 0.88 | 19.26 | 3.60 | 17.64 | 0.615 | 0.449 |
> | Llama-3.2-1B-Instruct (fine-tuned) | 11.32 | 39.60 | 18.67 | 34.94 | 0.814 | 0.610 |
> | DeepRTL-220m | 18.66 | 47.69 | 29.49 | 44.02 | 0.837 | 0.705 |
>
> | Generation (Syntax) | Success Rate | Pass@1 | Pass@5 |
> | --- | --- | --- | --- |
> | deepseek-coder-1.3b-instruct (original) | 44.52% | 12.90% | 67.74% |
> | deepseek-coder-1.3b-instruct (fine-tuned) | 60.00% | 38.71% | 77.42% |
> | Llama-3.2-1B-Instruct (original) | 45.16% | 12.90% | 70.97% |
> | Llama-3.2-1B-Instruct (fine-tuned) | 57.42% | 38.71% | 77.42% |
> | DeepRTL-220m | 78.06% | 70.97% | 80.65% |
>
> | Generation (Function) | Success Rate | Pass@1 | Pass@5 |
> | --- | --- | --- | --- |
> | deepseek-coder-1.3b-instruct (original) | 0% | 0% | 0% |
> | deepseek-coder-1.3b-instruct (fine-tuned) | 20.65% | 19.35% | 38.71% |
> | Llama-3.2-1B-Instruct (original) | 3.23% | 0.00% | 16.13% |
> | Llama-3.2-1B-Instruct (fine-tuned) | 21.94% | 19.35% | 45.16% |
> | DeepRTL-220m | 36.13% | 32.26% | 41.94% |
>
> [7] https://huggingface.co/deepseek-ai/deepseek-coder-1.3b-instruct
>
> [8] https://huggingface.co/meta-llama/Llama-3.2-1B-Instruct

---

> > ### Comment · Reviewer_UcZL · 2024-11-26
> > **great**
> >
> > good results. please include them into the revised manuscript

---

> > > ### Author Response · Authors · 2024-11-26
> > > **Revised Manuscript Uploaded**
> > >
> > > Thank you for your positive feedback on our results. We have uploaded the revised manuscript, which now includes the requested results. Additionally, we have addressed the feedback from all other reviewers and incorporated the necessary revisions throughout the manuscript.
> > >
> > > We hope the updated version meets your expectations. Please feel free to reach out if there are any further questions or clarifications needed.

---

> > > > ### Comment · Reviewer_vHqv · 2024-11-27
> > > > **Review adjustment**
> > > >
> > > > Thank you for the updated manuscript, and for the professional and constructive discussions here.
> > > > With the updates, I now feel comfortable enough to give this work a rating above the acceptance threshold, and will update my review accordingly.

---

> > > > > ### Author Response · Authors · 2024-11-27
> > > > > **Reply to Reviewer vHqv's Comment**
> > > > >
> > > > > Thank you for taking the time to review the updated manuscript and for considering a positive adjustment to your review score. We greatly appreciate your constructive feedback and are pleased to hear that the revisions have addressed your concerns.

---

> ### Comment · Reviewer_vHqv · 2024-11-25
> **Reponse to 1/4**
>
> Thank you for your detailed responses. I will respond in kind, under your four parts.
>
> tl;dr: I will adjust my score upwards based on your rebuttal; however, I still think there are significant adjustments that need to be adopted to make this "acceptable" to this conference. In particular:
> - Clarification of dataset release, and how proprietary data can be made available w/o violating licenses.
> - Weakening/adjustment of claims about vector similarity and GPT Score metrics.
> - Adjustment of "Progressive Learning" to bring it in line with established Curriculum Learning.
> - Inclusion of examples >2,048 tokens; or experiments in part 4/4 on same data as original model.
>
> **Challenges in Building a Foundation Model for Verilog** I do not dispute that building any sort of ML model for a low-resource language (Natural or Programming) is challenging, and in fact this is probably the number one factor why I think your *dataset* itself is a really nice contribution.
>
> **Main Contributions of Our Work**
> **1-3** I agree that the dataset should be very useful, and a lot of effort and thought went into its curation.\
> However (as other reviewers also pointed out), large -- and high-quality -- parts of this dataset are proprietary. You address this in one answer, to Reviewer hfVY (though not the others), saying that the dataset *including proprietary modules* will be released... how will that be possible, e.g., did you get licenses to publish these parts?
>
> **4** It doesn't "align with the principles of curriculum learning", it *is* curriculum learning.
>
> **5** These metrics are used on the descriptions of the code, i.e., on natural language (though even if it was on code directly similar approaches to code similarity with, for example, CodeBERT have been used before for evaluation as well as downstream tasks such as code clone detection or code retrieval).\
> I grant that this might be the first time these metrics are used in this way to explicitly evaluate Verilog code; however, the *metrics themselves* are far from novel.\
> I would let this pass if the claims were weaker, for example as you state yourself above: "To the best of our knowledge, this is the first application of these metrics to evaluate the code understanding capabilities of LLMs, providing a more robust and reliable assessment framework for code-learning tasks."\
> Though even in that case, I would have to question the assertion that BLEU and ROUGE are not well-suited for the evaluation task here, based on your own results. These metrics yield essentially the same evaluation results as embedding sim and GPT score. To show that there is merit in using these directly as metrics for evaluation, it should be shown that they correlate better with human judgements, or other "direct" automatic metrics such as code execution.

---

> ### Comment · Reviewer_vHqv · 2024-11-25
> **Reponse to 2/4**
>
> **1.** It is fair enough to say this hasn't been done for Verilog. Again, the part that "goes beyond simple fine-tuning" is the dataset, which I acknowledge is a very good contribution.
>
> **2.** Your "progressive learning" **is** curriculum learning.\
> I do concede this may be the first (or one of the first) times this is explicitly used for a code LM. [1] should be considered as contemporary. Still, this is an established term, and some background should be given about curriculum learning, and how it applies here.
>
> **3-4.** See answer to **5.** of your part 1/4.
>
> **5.** Conceded on the basis of this being primarily a dataset paper; still some re-work is required, especially regarding adjustments of claims wrt. evaluation metrics and curriculum learning.
>
> [1] Curriculum Learning for Small Code Language Models. Naïr et al., ACL2024.

---

> ### Comment · Reviewer_vHqv · 2024-11-25
> **Reponse to 3/4**
>
> **1.** This could have been a great opportunity to establish a benchmark that does in fact include longer examples, which would be an additional advantage over previous benchmarks. Table 3 shoes difficulties with complex designs; it is not obvious that "long" (in terms of tokens) equates to "complex" (in terms of semantics/task).
>
> **2.** Conceded.
>
> **3.** This seems not obvious from the manuscript, unless I missed a part where this is shown?
>
> **Choice of base model** Conceded.

---

> ### Comment · Reviewer_vHqv · 2024-11-25
> **Reponse to 4/4**
>
> These are great new additional experiments, and they should be included in an updated manuscript.
>
> However, if the claims in your previous answers about GPT-4o struggling with description generation for long modules hold, and this is used to generate the additional dataset examples over 2,048 tokens, these additional models should in fact **not** be trained on this data, as it would potentially (or even likely) introduce a lot of noise.\
> Instead, this should be trained on the exact same data as all other models, to make results comparable.

---

> > ### Author Response · Authors · 2024-11-26
> > **Reply to Reviewer vHqv's Response (Part 1/3)**
> >
> > Thank you for taking the time to provide a detailed response to our rebuttal and for considering an upward adjustment to your score. We deeply appreciate your constructive suggestions, and in the following, we will make further efforts to address your concerns.
> >
> > 1. **Clarification of Dataset Release:**
> >
> >     As mentioned in our response to Reviewer hfVY’s Q3, we plan to release all components of our work, including the full dataset (comprising both open-source and proprietary Verilog code along with their corresponding multi-level natural language descriptions), the Verilog understanding benchmark, and the model checkpoints, along with the training and evaluation scripts, soon after the paper is accpeted.
> >
> >     As detailed in Section 3.2 of the original manuscript, the open-source Verilog code constitutes the majority of our dataset, with 61,755 distinct Verilog modules, while the proprietary portion includes only 213 modules, derived from a set of purchased intellectual properties (IPs). We understand the importance of providing clear information regarding dataset release, particularly with respect to proprietary data and licensing restrictions. To address this, we have segmented the proprietary IPs into smaller modules and anonymized the data, ensuring that all datasets comply with the relevant licensing agreements and avoid any potential violations.
> >
> > 2. **Adjustment of Claims about Embedding Similarity and GPT Score Metrics:**
> >
> >     We acknowledge the need to adjust the claims regarding the novelty of the embedding similarity and GPT Score metrics to more accurately reflect their established use in other domains, like CodeBERT [1] for evaluating code similarities. While we believe that applying these metrics to Verilog understanding offers valuable insights, we recognize that they are not novel in the broader context of model evaluation. To address this, we will revise the manuscript to better align our claims with the established use of these metrics, emphasizing their role as complementary tools for evaluating semantic similarity in our specific context. Specifically, we will refrain from claiming that we propose these metrics. Instead, we will clarify that we are the first to apply them to evaluate the code understanding capabilities of LLMs, offering a more robust and reliable assessment framework for code-learning tasks.
> >
> >     **Effectiveness of BLEU and ROUGE**
> >
> >     In the original manuscript, we claim that embedding similarity and GPT score provide a more accurate assessment of semantic similarity between generated descriptions and ground truth summaries, compared to traditional metrics like BLEU and ROUGE, which are limited to surface-level n-gram overlaps. As shown in Table 2 and highlighted in Lines 471-475 of the original manuscript, BLEU and ROUGE yield inconsistent evaluations due to their inability to capture semantic meaning effectively. For example, while DeepRTL-16b excels in BLEU-4 and ROUGE-L, DeepRTL-220m performs better in ROUGE-1 and ROUGE-2. Similar inconsistencies arise when comparing GPT-3.5 and GPT-4, as well as in other cases. In contrast, embedding similarity and GPT score provide a more consistent and reliable evaluation of the models' abilities to understand Verilog code.
> >
> >     Additionally, we have conducted human evaluation, as detailed in Line 479-481 of the original manuscript, where DeepRTL-220m and GPT-4 achieve accuracies of 78% and 72%, respectively. To further highlight the limitations of BLEU and ROUGE, we also conducted human evaluation on o1-preview, which achieves an accuracy of 67%. These human evaluation results are in line with the findings from embedding similarity and GPT score metrics, but directly contradict the BLEU and ROUGE scores, which suggest that o1-preview outperforms GPT-4 in terms of Verilog understanding capabilities. Due to time constraints, we were only able to perform human evaluation for o1-preview, but we acknowledge that additional human evaluation is necessary to further demonstrate that embedding similarity and GPT score are more closely correlated with human judgments than traditional metrics.
> >
> >     [1] CodeBERT: A Pre-Trained Model for Programming and Natural Languages, EMNLP 2020.

---

> ### Author Response · Authors · 2024-11-26
> **Reply to Reviewer vHqv's Response (Part 2/3)**
>
> 3. **Adjustment of Progressive Training to Align with Curriculum Learning:**
>
>     We appreciate the feedback on aligning our terminology for "Progressive Training" with the broader concept of Curriculum Learning. In the revised manuscript, we will no longer use the term “Progressive Training”. Instead, we will explicitly state that we adapt curriculum learning principles to our specific setting for training unified models focused on Verilog understanding and generation. Additionally, we thank the reviewer for referencing a related work that applies curriculum learning to code language models [2]. We will include background information on this work and on curriculum learning in the Related Work section to further contextualize our approach.
>
>     [2] Curriculum Learning for Small Code Language Models, ACL2024.
>
> 4. **Experiments in Part 4/4 on Same Data as Original Model:**
>
>     Thank you for raising this important point. We agree that including dataset examples exceeding 2,048 tokens could introduce significant noise and reduce the comparability of the results. To address this, we have re-trained the two models discussed in Part 4/4 using the same dataset as DeepRTL and present the updated results in the following tables. Notably, after excluding examples longer than 2,048 tokens, the performance of the fine-tuned models for both Verilog understanding and generation shows significant improvements.
>
>     | Understanding | BLEU-4 | ROUGE-1 | ROUGE-2 | ROUGE-L | Emb. Sim. | GPT Score |
>     | --- | --- | --- | --- | --- | --- | --- |
>     | deepseek-coder-1.3b-instruct (original) | 1.04 | 21.43 | 4.38 | 19.77 | 0.678 | 0.557 |
>     | deepseek-coder-1.3b-instruct (fine-tuned with same data) | 11.96 | 40.49 | 19.77 | 36.14 | 0.826 | 0.664 |
>     | Llama-3.2-1B-Instruct (original) | 0.88 | 19.26 | 3.60 | 17.64 | 0.615 | 0.449 |
>     | Llama-3.2-1B-Instruct (fine-tuned with same data) | 12.11 | 39.95 | 19.47 | 35.29 | 0.825 | 0.620 |
>     | DeepRTL-220m | 18.66 | 47.69 | 29.49 | 44.02 | 0.837 | 0.705 |
>
>     | Generation (Syntax) | Success Rate | Pass@1 | Pass@5 |
>     | --- | --- | --- | --- |
>     | deepseek-coder-1.3b-instruct (original) | 44.52% | 12.90% | 67.74% |
>     | deepseek-coder-1.3b-instruct (fine-tuned with same data) | 63.87% | 61.29% | 80.65% |
>     | Llama-3.2-1B-Instruct (original) | 45.16% | 12.90% | 70.97% |
>     | Llama-3.2-1B-Instruct (fine-tuned with same data) | 58.71% | 54.84% | 80.65% |
>     | DeepRTL-220m | 78.06% | 70.97% | 80.65% |
>
>     | Generation (Function) | Success Rate | Pass@1 | Pass@5 |
>     | --- | --- | --- | --- |
>     | deepseek-coder-1.3b-instruct (original) | 0% | 0% | 0% |
>     | deepseek-coder-1.3b-instruct (fine-tuned with same data) | 25.81% | 22.58% | 48.39% |
>     | Llama-3.2-1B-Instruct (original) | 3.23% | 0.00% | 16.13% |
>     | Llama-3.2-1B-Instruct (fine-tuned with same data) | 22.58% | 19.35% | 48.39% |
>     | DeepRTL-220m | 36.13% | 32.26% | 41.94% |
> 5. **“A Great Opportunity to Establish A Benchmark Containing Longer Examples” & “LLMs Struggle with Complex Designs”:**
>
>     Thank you for your thoughtful feedback. We agree that developling a Verilog generation benchmark with longer examples is important. However, this is a non-trivial task, as it requires establishing a testbench for each sample to assess the functional accuracy of the generated designs. Currently, there is no automated approach to generate these testbenches. Nevertheless, we recognize the value of this direction and, in future work, we plan to explore the development and evaluation of LLMs capable of handling longer Verilog designs. This could involve dedicating additional efforts to building a new benchmark with longer examples.
>
>     Additionally, we also think that longer designs do not necessarily equate to more complex designs. As noted in our previous response in Part 3/4, current LLMs are often limited to generating simpler designs and struggle with more complex ones. For instance, as shown in Table 3 of the original manuscript, almost all evaluated models can generate the *adder_8bit* design correctly. However, when it comes to generating the *adder_32bit* and *adder_64bit* designs, all models fail to produce functionally correct results. Furthermore, it inspires us that establishing a metric to assess the complexity of Verilog designs would provide valuable insights into model performance, and we plan to consider this in future work.

---

> ### Author Response · Authors · 2024-11-26
> **Reply to Reviewer vHqv's Response (Part 3/3)**
>
> 6. **Clarification on Existing LLMs Struggling with Correctness when Generating Descriptions for Longer Designs:**
>
>     We apologize for any confusion regarding this point. The observation about the accuracy of descriptions generated for longer designs is based on empirical findings from our experimentation process. Since we rely on GPT-4 to generate descriptions for our dataset, ensuring the correctness of these descriptions is critical. To address this, we have conducted multiple rounds of description generation followed by human evaluation.
>
>     During the annotation process, we divided the dataset into two sections: Verilog designs with fewer than 2,048 tokens, and designs with token lengths between 2,048 and 4,096 tokens. Our human evaluation finds that descriptions for Verilog designs with fewer than 2,048 tokens are approximately 90% accurate, while descriptions for designs with token lengths between 2,048 and 4,096 tokens have accuracy rates of only 60%–70%. And accuracy further decreases for designs exceeding 4,096 tokens.
>
>     In Line 160-161 of the original manuscript, we state that "This segmentation is crucial given the limited context length of current LLMs, improving the efficiency and accuracy of the subsequent annotation and fine-tuning processes." We acknowledge that this may have been unclear to readers, and we will provide further clarification in the revised manuscript to ensure the explanation is more explicit.
>
>     Additionally, through our experiments with fine-tuning deepseek-coder-1.3b-instruct and Llama-3.2-1B-Instruct—both with and without Verilog designs exceeding 2,048 tokens—we further demonstrate that existing LLMs struggle with generating accurate descriptions for longer designs. These longer examples introduce significant noise, which negatively impacts the model’s performance.
>
>
> We sincerely appreciate your thoughtful feedback, which has highlighted important areas for further refinement. We are fully committed to addressing these points in the revised manuscript and believe these adjustments will enhance both the quality and impact of our work. Please feel free to share any additional feedback if necessary and we highly value your insights.

---

### Official Review · Reviewer_Aczy · 2024-10-30

**Soundness:** 2
**Presentation:** 3
**Contribution:** 3
**Rating:** 8
**Confidence:** 4

**Summary:**

The paper makes a contribution to the field of hardware design automation by addressing both the generation and understanding of Verilog code using large language models (LLMs). While previous studies primarily focused on the generation aspect, this work recognizes the importance of understanding Verilog code and proposes a unified representation model, DeepRTL, built on an enhanced CodeT5+ architecture. This model is trained on a specifically curated dataset that tightly aligns natural language descriptions with Verilog code, aiming to improve the semantic alignment between the two. Additionally, the paper introduces the first benchmark specifically for Verilog understanding and develops two novel metrics, embedding similarity and GPT score, to capture semantic similarities more effectively than traditional n-gram-based metrics like BLEU and ROUGE. In comparative assessments, DeepRTL surpasses GPT-4 in Verilog understanding tasks and matches the performance of OpenAI’s o1-preview model in code generation tasks.

**Strengths:**

1. The paper introduces a novel task for evaluating LLMs in hardware design, focusing on Verilog understanding—prior work mainly focuses on generation. It introduces new training datasets, evaluation benchmarks, and establishes baselines for this new task.

2. DeepRTL, the model proposed in this paper, uniquely good at both the generation and understanding of Verilog, making it different from other models in the hardware design domain.

3. The methodology for creating a natural language-code parallel corpus via prompt engineering with GPT-4 is innovative and shows promise for broader application in fields where parallel corpora are lacking.

4. The diagrams in this paper describes the proposed methods clearly and intuitively.

**Weaknesses:**

1. The reason for selecting T5-like models as the base for DeepRTL is not empirically validated. It remains unclear whether the observed performance gains in Verilog understanding are due to T5's encoder-decoder architecture or the synthesized dataset used for fine-tuning. Comparative analysis with a decoder-only model, such as LLaMa-3-1B or DeepSeekCoder-1.3B, using the same dataset for finetuning would provide clearer insights.

2. The paper does not evaluate the impact of varying context window lengths, which is important given that CodeT5+ supports a limited token count (2,048 tokens), while actual Verilog code often exceeds this length. Dropping examples longer than 2,048 tokens will also bias the results in favor of DeepRTL, which is based on CodeT5+. A model accommodating longer context windows could potentially offer superior performance on the general task, but not for this tailored dataset.

3. The evaluation metrics for code understanding—embedding similarity and GPT score—are solely based on GPT models, leading to potential bias, as evidenced by the inflated scores of GPT-3.5, GPT-4, and o1-preview models shown in Table 2. This overlap may make the comparisons bias in favor of GPT-family models.

4. The evaluation of code generation lacks a comprehensive set of baselines. Despite mentioning various Verilog generation models in the related work section, these models are absent from the comparative analysis in Table 3.

5. The fine-tuning dataset includes proprietary code that cannot be released publicly, and the benchmarks used are also developed by the authors. The absence of shared code, data, or models in the publication hinders reproducibility and make it impossible to assess potential data contamination and bias in evaluation.

**Questions:**

N/A

---

> ### Author Response · Authors · 2024-11-25
> **Response to Reviewer Aczy (Part 1/4)**
>
> ### Q1: “The reason for selecting T5-like models as the base for DeepRTL is not empirically validated. It remains unclear whether the observed performance gains in Verilog understanding are due to T5's encoder-decoder architecture or the synthesized dataset used for fine-tuning. Comparative analysis with a decoder-only model, such as LLaMa-3-1B or DeepSeekCoder-1.3B, using the same dataset for finetuning would provide clearer insights.”
>
> R1: We thank the reviewer for this insightful feedback. In this work, we choose CodeT5+, a family of encoder-decoder code foundation models, as the base model for training DeepRTL for two primary reasons. First, as we aim to develop a unified model for Verilog understanding and generation, T5-like models are particularly well-suited due to their ability to effectively handle both tasks, as evidenced by [1]. Second, the encoder component of CodeT5+ enables the natural extraction of Verilog representations, which can be potentially utilized for various downstream tasks in Electronic Design Automation (EDA) at the RTL stage. Examples include PPA (Power, Performance, Area) prediction, which estimates the power consumption, performance, and area of an RTL design, and verification, which ensures that the RTL design correctly implements its intended functionality and meets specification requirements, which are two critical tasks in the hardware design process. This capability distinguishes it from decoder-only models, which are typically less suited for producing standalone, reusable intermediate representations. In future work, we plan to explore how DeepRTL can further enhance productivity in the hardware design process.
>
> **Comparative Analysis with Decoder-Only Models**
>
> To further demonstrate the superiority of CodeT5+ as a base model, we fine-tune two additional models, deepseek-coder-1.3b-instruct [2] and Llama-3.2-1B-Instruct [3], using our proposed dataset and progressive training strategy. Notably, the maximum input length for deepseek-coder-1.3b-instruct is 16k tokens, and for Llama-3.2-1B-Instruct, it is 128k tokens. As a result, we did not exclude Verilog modules exceeding 2,048 tokens in these two cases.
>
> In the following tables, we present the performance of both the original base models and their fine-tuned counterparts on Verilog understanding and generation tasks, alongside the results from our DeepRTL-220m model. The improvement in performance from the original base models to the fine-tuned models highlights the effectiveness of our dataset and progressive fine-tuning strategy. Meanwhile, the superior performance of DeepRTL-220m on both tasks, despite its smaller model size, underscores the architectural advantages of our approach.
>
> We hope these experimental results can provide more insights into the impact of model architecture, as well as the influence of our proposed training dataset and strategy, on final performance. These experimental results will be incorporated into the revised manuscript.
>
> | Understanding | BLEU-4 | ROUGE-1 | ROUGE-2 | ROUGE-L | Emb. Sim. | GPT Score |
> | --- | --- | --- | --- | --- | --- | --- |
> | deepseek-coder-1.3b-instruct (original) | 1.04 | 21.43 | 4.38 | 19.77 | 0.678 | 0.557 |
> | deepseek-coder-1.3b-instruct (fine-tuned) | 11.27 | 40.28 | 18.95 | 35.93 | 0.825 | 0.649 |
> | Llama-3.2-1B-Instruct (original) | 0.88 | 19.26 | 3.60 | 17.64 | 0.615 | 0.449 |
> | Llama-3.2-1B-Instruct (fine-tuned) | 11.32 | 39.60 | 18.67 | 34.94 | 0.814 | 0.610 |
> | DeepRTL-220m | 18.66 | 47.69 | 29.49 | 44.02 | 0.837 | 0.705 |
>
> | Generation (Syntax) | Success Rate | Pass@1 | Pass@5 |
> | --- | --- | --- | --- |
> | deepseek-coder-1.3b-instruct (original) | 44.52% | 12.90% | 67.74% |
> | deepseek-coder-1.3b-instruct (fine-tuned) | 60.00% | 38.71% | 77.42% |
> | Llama-3.2-1B-Instruct (original) | 45.16% | 12.90% | 70.97% |
> | Llama-3.2-1B-Instruct (fine-tuned) | 57.42% | 38.71% | 77.42% |
> | DeepRTL-220m | 78.06% | 70.97% | 80.65% |
>
> | Generation (Function) | Success Rate | Pass@1 | Pass@5 |
> | --- | --- | --- | --- |
> | deepseek-coder-1.3b-instruct (original) | 0% | 0% | 0% |
> | deepseek-coder-1.3b-instruct (fine-tuned) | 20.65% | 19.35% | 38.71% |
> | Llama-3.2-1B-Instruct (original) | 3.23% | 0.00% | 16.13% |
> | Llama-3.2-1B-Instruct (fine-tuned) | 21.94% | 19.35% | 45.16% |
> | DeepRTL-220m | 36.13% | 32.26% | 41.94% |
>
> [1] CodeT5+: Open Code Large Language Models for Code Understanding and Generation. EMNLP 2023.
>
> [2] https://huggingface.co/deepseek-ai/deepseek-coder-1.3b-instruct
>
> [3] https://huggingface.co/meta-llama/Llama-3.2-1B-Instruct

---

> ### Author Response · Authors · 2024-11-25
> **Response to Reviewer Aczy (Part 2/4)**
>
> ### Q2: “The paper does not evaluate the impact of varying context window lengths, which is important given that CodeT5+ supports a limited token count (2,048 tokens), while actual Verilog code often exceeds this length. Dropping examples longer than 2,048 tokens will also bias the results in favor of DeepRTL, which is based on CodeT5+. A model accommodating longer context windows could potentially offer superior performance on the general task, but not for this tailored dataset.”
>
> R2: Thank you for your valuable and thoughtful feedback. In this work, we exclude Verilog modules exceeding 2,048 tokens for reasons beyond the maximum context length limitation of our base model, CodeT5+:
>
> 1. **Generation Capabilities of Existing LLMs Are Limited to Small Designs**
>
>     The existing benchmarks for Verilog generation, including the one used in our work [4], do not include designs exceeding 2,048 tokens. The maximum token length observed in the benchmark is 1,851. As shown in Table 3 of the original manuscript, even the state-of-the-art LLM, o1-preview, is limited to generating simple designs accurately and lacks the capability to handle more complex designs. To clarify why we exclude Verilog modules beyond 2,048 tokens, we will include a figure in the revised manuscript that shows the token length distribution across the benchmark.
>
>     We recognize the importance of evaluating models on Verilog code exceeding 2,048 tokens, as real-world Verilog designs often surpass this threshold. However, creating a new benchmark tailored for longer examples presents significant challenges, particularly due to the lack of automated tools for generating testbenches for these extended cases.
>
> 2. **Segmentation as a Common Practice**
>
>     Segmenting longer code into smaller chunks that fit within the predefined context window and discarding those that exceed it is a widely accepted practice in both Verilog-related research ([5] and [6]) and studies on software programming languages [1]. This approach ensures compatibility with current LLMs while maintaining the integrity and usability of the dataset. It is worth noting that the default maximum sequence length in CodeT5+ is 512 tokens, and our work extends this limit to 2,048 tokens to better accommodate Verilog designs.
>
> 3. **Empirical Findings and Practical Challenges**
>
>     Our experiments reveal a key empirical observation: existing LLMs, such as GPT-4, consistently produce accurate descriptions for shorter Verilog modules but struggle with correctness when handling longer ones. Since our datasets rely on LLM-generated annotations, restricting the dataset to Verilog modules within the 2,048-token limit helps maintain the quality and accuracy of annotations. This, in turn, facilitates higher-quality dataset creation and more efficient fine-tuning.
>
>
> **Additional Experiments to Examine the Impact of Varying Context Window Lengths**
>
> To investigate the impact of context window length, we exclude all Verilog modules exceeding 512 tokens and use the truncated dataset to train a new model, DeepRTL-220m-512, with a maximum input length of 512 tokens and using our progressive training strategy. We then evaluate both DeepRTL-220m-512 and DeepRTL-220m on the Verilog understanding benchmark samples, where the length of the modules is below 512 tokens, and present the results in the following table. For the generation task, DeepRTL-220m-512 demonstrates near-zero performance, with almost 0% accuracy for both syntax and functional correctness. This result challenges the statement, “A model accommodating longer context windows could potentially offer superior performance on the general task, but not for this tailored dataset,” as it does not hold true in this case.
>
> | Understanding (samples below 512 tokens) | BLEU-4 | ROUGE-1 | ROUGE-2 | ROUGE-L | Emb. Sim. | GPT-Score |
> | --- | --- | --- | --- | --- | --- | --- |
> | DeepRTL-220m-512 | 14.98 | 44.27 | 23.11 | 40.08 | 0.780 | 0.567 |
> | DeepRTL-220m | 18.74 | 48.41 | 29.82 | 45.01 | 0.855 | 0.743 |
>
> Together with our response to Q1, we hope to provide further insights into the influence of context window length on model performance. These experimental results will be included in the updated manuscript.
>
> [4] Natural language is not enough: Benchmarking multi-modal generative AI for Verilog generation, ICCAD 2024.
>
> [5] Data is all you need: Finetuning LLMs for Chip Design via an Automated design-data augmentation framework. DAC 2024.
>
> [6] BetterV: Controlled Verilog Generation with Discriminative Guidance. ICML 2024.

---

> ### Author Response · Authors · 2024-11-25
> **Response to Reviewer Aczy (Part 3/4)**
>
> ### Q3: “The evaluation metrics for code understanding—embedding similarity and GPT score—are solely based on GPT models, leading to potential bias, as evidenced by the inflated scores of GPT-3.5, GPT-4, and o1-preview models shown in Table 2. This overlap may make the comparisons bias in favor of GPT-family models.”
>
> R3: Thank you for this insightful feedback. In this work, we introduce two evaluation metrics, embedding similarity and GPT score, for evaluating Verilog understanding, as they can better capture the semantic similarity between generated descriptions and ground truth summaries. Traditional metrics such as BLEU and ROUGE primarily focus on lexical similarities and often fail to accurately reflect semantic nuances, which are critical for evaluating code understanding tasks.
>
> We select GPT models to compute embedding similarity and GPT score because they represent the most powerful general-purpose LLMs currently available. Their advanced capabilities allow for more nuanced and semantically rich evaluations, which we believe enhance the accuracy and reliability of these metrics. However, as the reviewer rightly points out, this approach may introduce a potential bias in favor of GPT-family models, given that these metrics are derived from the same class of models. We also recognize that some uncertainty may exist in these metrics due to the reliance on GPT-generated representations and scores.
>
> To mitigate potential biases and provide a more comprehensive assessment, we complement these automated metrics with human evaluation. As detailed in the original manuscript Line 479-480, our human evaluation results demonstrate that DeepRTL-220m and GPT-4 achieve accuracies of 78% and 72%, respectively. This independent validation highlights the robustness of DeepRTL’s understanding capabilities, even when compared against a strong baseline like GPT-4.
>
> ### Q4: “The evaluation of code generation lacks a comprehensive set of baselines. Despite mentioning various Verilog generation models in the related work section, these models are absent from the comparative analysis in Table 3.”
>
> R4: Thank you for this constructive feedback. In this work, we choose OpenAI’s GPT-3.5, GPT-4, and o1-preview as baseline models for comparison. These models represent the most advanced general-purpose LLMs currently available, with demonstrated excellence across various domains, including Verilog generation [5][7][8]. Notably, o1-preview is the latest model specifically designed to handle complex reasoning and coding tasks [9], and it achieves superior performance in Verilog generation in our experiments.
>
> To further show the superiority of DeepRTL, we conduct experiments comparing it with models specifically trained on Verilog. We did not select Zhang et al., 2024 [10] and Chang et al., 2024b [5] for comparison, as their models are not open-sourced, and it is non-trivial to reproduce their experiments. Additionally, the reported performance in their original papers is either comparable to, and in some cases inferior to, that of GPT-3.5. In the following tables, we compare two state-of-the-art Verilog generation models, RTLCoder-Deepseek-v1.1 [11] and fine-tuned-codegen-16B-Verilog [12] with our DeepRTL-220m. Notably, RTLCoder-Deepseek-v1.1 is fine-tuned on DeepSeek-coder-6.7b, and fine-tuned-codegen-16B-Verilog is fine-tuned on CodeGen-multi-16B, both of which have significantly larger parameter sizes than DeepRTL-220m. Despite this, the superior performance of DeepRTL-220m further demonstrates the effectiveness of our proposed dataset and progressive training strategy. And we will incorporate these experimental results in the updated manuscript.
>
> | Understanding | BLEU-4 | ROUGE-1 | ROUGE-2 | ROUGE-L | Emb. Sim. | GPT Score |
> | --- | --- | --- | --- | --- | --- | --- |
> | RTLCoder-Deepseek-v1.1 | 1.08 | 21.83 | 4.68 | 20.30 | 0.687 | 0.561 |
> | fine-tuned-codegen-16B-Verilog | 0.09 | 6.54 | 0.35 | 6.08 | 0.505 | 0.311 |
> | DeepRTL-220m | 18.66 | 47.69 | 29.49 | 44.02 | 0.837 | 0.705 |
>
> | Generation (syntax) | Success Rate | Pass@1 | Pass@5 |
> | --- | --- | --- | --- |
> | RTLCoder-Deepseek-v1.1 | 48.39% | 41.94% | 77.42% |
> | fine-tuned-codegen-16B-Verilog | 50.97% | 48.39% | 70.97% |
> | DeepRTL-220m | 78.06% | 70.97% | 80.65% |
>
> | Generation (function) | Success Rate | Pass@1 | Pass@5 |
> | --- | --- | --- | --- |
> | RTLCoder-Deepseek-v1.1 | 20.00% | 16.13% | 35.48% |
> | fine-tuned-codegen-16B-Verilog | 12.26% | 9.68% | 32.26% |
> | DeepRTL-220m | 36.13% | 32.26% | 41.94% |
>
> [7] Verigen: A large language model for verilog code generation. TODAES 2024.
>
> [8] RTLCoder: Fully Open-Source and Efficient LLM-Assisted RTL Code Generation Technique. TCAD 2024.
>
> [9] https://openai.com/index/introducing-openai-o1-preview/
>
> [10] MG-Verilog: Multi-grained Dataset Towards Enhanced LLM-assisted Verilog Generation. LAD 2024.
>
> [11] https://huggingface.co/ishorn5/RTLCoder-Deepseek-v1.1
>
> [12] https://huggingface.co/shailja/fine-tuned-codegen-16B-Verilog

---

> ### Author Response · Authors · 2024-11-25
> **Response to Reviewer Aczy (Part 4/4)**
>
> ### Q5: “The fine-tuning dataset includes proprietary code that cannot be released publicly, and the benchmarks used are also developed by the authors. The absence of shared code, data, or models in the publication hinders reproducibility and make it impossible to assess potential data contamination and bias in evaluation.”
>
> R5: Thank you for raising this point. We plan to release all components of our work soon following the acceptance of the paper. This includes the complete dataset (comprising both open-source and proprietary Verilog code with their corresponding multi-level natural language descriptions), the Verilog understanding benchmark, model checkpoints, as well as the training and evaluation scripts.

---

> > ### Comment · Reviewer_Aczy · 2024-11-25
> >
> > Thank you for your detailed response and new experimental results. I decide to increase my review score from 6 to 8.

---

> > > ### Author Response · Authors · 2024-11-27
> > > **Response to Reviewer Aczy's Comment**
> > >
> > > Thank you for your thoughtful reconsideration of our work and for taking the time to review the additional experimental results. We greatly appreciate your decision to adjust the review score upward and are pleased that our responses have addressed your concerns.
> > >
> > > Additionally, for Q1, as Reviewer vHqv points out, including dataset examples exceeding 2,048 tokens when fine-tuning deepseek-coder-1.3b-instruct and Llama-3.2-1B-Instruct could introduce significant noise and reduce the comparability of the results. To address this, we have re-trained these two models using the same dataset as DeepRTL and present the updated results in the following tables. Notably, after excluding examples longer than 2,048 tokens, the performance of the fine-tuned models for both Verilog understanding and generation shows significant improvements. This further supports our rationale for excluding examples longer than 2,048 tokens. We hope these updated results could provide further clarity and more insights for you, and we have incorporated all of these experiments into the revised manuscript.
> > >
> > > | Understanding | BLEU-4 | ROUGE-1 | ROUGE-2 | ROUGE-L | Emb. Sim. | GPT Score |
> > > | --- | --- | --- | --- | --- | --- | --- |
> > > | deepseek-coder-1.3b-instruct (original) | 1.04 | 21.43 | 4.38 | 19.77 | 0.678 | 0.557 |
> > > | deepseek-coder-1.3b-instruct (fine-tuned with same data) | 11.96 | 40.49 | 19.77 | 36.14 | 0.826 | 0.664 |
> > > | Llama-3.2-1B-Instruct (original) | 0.88 | 19.26 | 3.60 | 17.64 | 0.615 | 0.449 |
> > > | Llama-3.2-1B-Instruct (fine-tuned with same data) | 12.11 | 39.95 | 19.47 | 35.29 | 0.825 | 0.620 |
> > > | DeepRTL-220m | 18.66 | 47.69 | 29.49 | 44.02 | 0.837 | 0.705 |
> > >
> > > | Generation (Syntax) | Success Rate | Pass@1 | Pass@5 |
> > > | --- | --- | --- | --- |
> > > | deepseek-coder-1.3b-instruct (original) | 44.52% | 12.90% | 67.74% |
> > > | deepseek-coder-1.3b-instruct (fine-tuned with same data) | 63.87% | 61.29% | 80.65% |
> > > | Llama-3.2-1B-Instruct (original) | 45.16% | 12.90% | 70.97% |
> > > | Llama-3.2-1B-Instruct (fine-tuned with same data) | 58.71% | 54.84% | 80.65% |
> > > | DeepRTL-220m | 78.06% | 70.97% | 80.65% |
> > >
> > > | Generation (Function) | Success Rate | Pass@1 | Pass@5 |
> > > | --- | --- | --- | --- |
> > > | deepseek-coder-1.3b-instruct (original) | 0% | 0% | 0% |
> > > | deepseek-coder-1.3b-instruct (fine-tuned with same data) | 25.81% | 22.58% | 48.39% |
> > > | Llama-3.2-1B-Instruct (original) | 3.23% | 0.00% | 16.13% |
> > > | Llama-3.2-1B-Instruct (fine-tuned with same data) | 22.58% | 19.35% | 48.39% |
> > > | DeepRTL-220m | 36.13% | 32.26% | 41.94% |

---

### Official Review · Reviewer_UcZL · 2024-11-05

**Soundness:** 3
**Presentation:** 2
**Contribution:** 2
**Rating:** 6
**Confidence:** 3

**Summary:**

This paper deals with the task of code understanding and generation in the context of generation of hardware description language (HDL) code. In particular, this work focuses on the generation of Verilog code.
The model is based on an existing CodeLLM (authors used CodeT5+), which was fine tuned with a new augmented dataset created for this purpose. The dataset comprises both open and proprietary verilog codes, which were augmented (commented and summarised) by GPT-4.
Two models are trained using a  progressive training strategy based on CodeT5+ models. For the understanding benchmark, models are evaluated in terms of BLUE and ROUGE, as well as embedding similarity and GPT score. Results show an improved performance over competitors and baseline models. For the generation part, the models are evaluated on a Verilog generation benchmark introduced by Chang et al. 2024, and compared with GPT series models showing competitive performance against the best, o1-preview and surpassing GPT3.5 and GPT4.

**Strengths:**

Original approach, focusing on both generation and understanding tasks on a low resource code language as Verilog, specifically designed for hardware description.
The approach seems reasonable. The field of application is needed and follows the ultimate goal of improving electronic design automation.

**Weaknesses:**

The work lacks clarity. Particularly, the dataset collection and the training regime are not completely clear and their figures do not clarify the issue (see below).
Experiments seem reasonable but all baselines and competitors weren’t trained specifically on verilog. Since the current work cites other previous approaches, experiments could have compared to them as well (or explain why was not possible)

**Questions:**

* Verilog is not a particularly known language. Authors could have explained a bit more its nature, syntax and usage.

* Figure 1, although it helps to understand the flow of data collection, it’s not particularly clear. The fact that the flow goes to the top-left in opposition to the common flow for reading (top to bottom and left to right) makes it unclear. Also, which part is used for training? Only after distil?

* Line 388-392: these lines and Figure 3 describe the progressive training. This explanation is not clear. Are the authors just feeding the model with more to less granular annotations? That could be an example of Curriculum learning. Please clarify and add references if needed.

* why the authors didn’t compared the performance of the new models with Zhang et al., 2024, , Chang et al. 2024b, Liu et al. (2023b); Thakur et al. (2024),

**Details Of Ethics Concerns:**

--

---

> ### Author Response · Authors · 2024-11-25
> **Response to Reviewer UcZL (Part 1/3)**
>
> ### Q1: “Verilog is not a particularly known language. Authors could have explained a bit more about its nature, syntax and usage.”
>
> R1: Thank you for pointing this out. Verilog is the most widely used hardware description language (HDL) for modeling and designing digital circuits. While software programming languages like Python or C++ are used to write instructions that control a computer’s CPU, Verilog defines the structure and behavior of hardware systems such as processors and memory. Below, we outline some key characteristics and syntax of Verilog:
>
> 1. **Modules and Hierarchy:** Verilog’s primary building blocks are **modules**, analogous to functions or classes in programming languages like Python or C++. A Verilog module defines a unit of hardware that can represent anything from a simple gate to a complete processor. Each module in Verilog encapsulates inputs, outputs, and internal logic, and modules can be instantiated within other modules, enabling hierarchical designs that mirror the complexity of real-world systems.
> 2. **Concurrent Execution:** A defining feature of Verilog, and a key difference from software programming languages, is its inherent **concurrency**. Hardware systems operate in parallel, and Verilog models this behavior using constructs such as `always` blocks and `assign` statements. In contrast, software languages like Python typically execute instructions sequentially (line-by-line).
> 3. **Time-Driven Behavior:** Verilog programs are time-sensitive and often use constructs like **delays** (`#`), **timing controls**, and **clock-driven processes** to model the behavior of hardware over time. The `always` and `initial` blocks define how signals evolve, enabling precise descriptions of the temporal dynamics crucial to digital systems.
> 4. **Control Flow and Data Types:** Verilog supports familiar **control structures** (e.g., `if`, `else`, `for` loops) and **data types** (e.g., integers, registers, and wires), but these are adapted to represent hardware signals. For instance, `wire` represents a connection between components, while `reg` is used to store values, distinguishing them from variables in software programming.
>
> Verilog is used extensively for designing digital circuits at various levels of abstraction, from high-level functional descriptions to low-level gate-level representations. It is employed in simulation, synthesis, and verification tasks to ensure that a design behaves as expected before it is physically implemented on hardware.
>
> We recognize that the lack of explanations about Verilog might confuse readers unfamiliar with the language. We will add a section to introduce Verilog in the revised manuscript. Additionally, we refer the reviewer to our response to Reviewer hfVY’s Q1 for more information.
>
> ### Q2: “Figure 1 is not particularly clear. The fact that the flow goes to the top-left in opposition to the common flow for reading (top to bottom and left to right) makes it unclear. Also, which part is used for training? Only after distil?”
>
> R2: Thank you for your valuable feedback regarding Figure 1. We appreciate your observation about the flow direction and its potential impact on clarity. In the revised manuscript, we will update Figure 1 to ensure the flow follows the conventional reading direction (top to bottom and left to right), making it more intuitive and easier to follow.
>
> **Clarification on Data Used for Training:** We apologize for the confusion regarding which parts of the data are used for training. To clarify, our entire dataset is utilized during training. Specifically, the data from the *Line Comment*, *Specification*, and *Functional Description* blocks in Figure 1 are all included in the training process.
>
> For further context, Figure 2 provides a comprehensive example of our annotation process for a complete Verilog module. This example illustrates three levels of annotation: **line**, **block**, and **module**, with each level containing descriptions that span various levels of detail—from detailed specifications to high-level functional descriptions. All these annotations, across all levels and degrees of detail, are fully used in the training process. Additionally, Table 1 in the original manuscript summarizes the overall statistics of the training data.
>
> We acknowledge that this was not made sufficiently clear in the original manuscript. In the revised version, we will explicitly indicate which parts of the dataset are used for training to avoid any ambiguity.

---

> ### Author Response · Authors · 2024-11-25
> **Response to Reviewer UcZL (Part 2/3)**
>
> ### Q3: “The explanation for the progressive training is not clear. Are the authors just feeding the model with more to less granular annotations? That could be an example of curriculum learning. Please clarify and add references if needed.”
>
> R3: Thank you for your valuable feedback. As noted in our response to Q2, our dataset includes three levels of annotation: **line**, **block**, and **module**, with each level containing descriptions that span various levels of detail—from detailed specifications to high-level functional descriptions. And the entire dataset is utilized for training. To fully leverage the potential of this dataset, we employ a **progressive training strategy**, enabling the model to incrementally build knowledge by starting with simpler cases and advancing to more complex ones.
>
> The progressive training strategy involves transitioning from more granular to less granular annotations across hierarchical levels, which can be conceptualized as a **tree structure** with the following components:
>
> 1. **Hierarchical Levels (Tree Root):** The training process transitions sequentially across the three hierarchical levels—**line**, **block**, and **module**. Each level is fully trained before moving to the next, ensuring a solid foundation at simpler levels before addressing more complex tasks.
> 2. **Granularity of Descriptions (Second Layer):** Within each hierarchical level, the annotations transition from **detailed descriptions** to **high-level descriptions**. This progression ensures that the model learns finer details first and then builds an understanding of higher-level abstractions.
> 3. **Annotation Source Transition (Third Layer):** At each level and granularity, training starts with **GPT-annotated data** and is followed by **human-annotated data**. This sequence leverages large-scale machine-generated annotations first and refines the model with high-quality, human-curated data.
> 4. **Instruction Blending**: Each terminal node in this tree represents a specific training dataset, which blends tasks for **Verilog understanding** and **Verilog generation**. This enables the model to perform well across diverse tasks.
>
> The training process mirrors a **pre-order traversal** of this tree structure:
>
> 1. Starting at the root, training begins with the **line level**.
> 2. The model progresses through the second layer (detailed, medium-detail, and high-level descriptions).
> 3. Within each granularity, training transitions through the third layer (GPT-annotated data first, followed by human-annotated data).
> 4. Once the line level is complete, the process repeats for the **block level** and then the **module level**.
>
> This progressive training strategy aligns closely with the principles of **curriculum learning**, where simpler concepts are introduced first, and the knowledge gained is transferred incrementally to handle more complex scenarios.
>
> To validate the effectiveness of this strategy, we conducted an ablation study where the model was trained on the entire dataset all at once without progression. The results, presented in Table 2 of the original manuscript, demonstrate that the progressive training strategy significantly outperforms this baseline approach. Moreover, to the best of our knowledge, this is the first application of a curriculum-like training strategy in the code-learning domain. Unlike existing Verilog-related models that establish simple and weak alignments between natural language and Verilog code [1], or general software code datasets like CodeSearchNet [2] that only provide single-level docstring annotations, our approach incorporates multi-level and multi-granularity annotations in a structured training process.
>
> We acknowledge that the explanation of the progressive training strategy in the original manuscript has lacked clarity. In the revised manuscript, we will enhance **Section 4.3** to provide a more detailed explanation and improve **Figure 3** to better illustrate this process. Specifically, we will include a tree-like figure to visualize the hierarchical training structure, which we believe will make the strategy clearer and more intuitive.
>
> [1] Data is all you need: Finetuning LLMs for Chip Design via an Automated design-data augmentation framework, DAC 2024
>
> [2] https://huggingface.co/datasets/code-search-net/code_search_net

---

> ### Author Response · Authors · 2024-11-25
> **Response to Reviewer UcZL (Part 3/3)**
>
> ### Q4: “Experiments seem reasonable but all baselines and competitors weren’t trained specifically on verilog.” & “Why the authors didn’t compare the performance of the new models with Zhang et al., 2024; Chang et al., 2024b; Liu et al., 2023b; Thakur et al., 2024.?”
>
> R4: Thank you for your thoughtful feedback. In this work, we choose OpenAI’s GPT-3.5, GPT-4, and o1-preview as baseline models for comparison. Notably, o1-preview is the latest model designed to solve complex tasks, including coding [3], and demonstrates superior performance in Verilog generation in our experiments. While it is true that these models are not specifically trained on Verilog, they represent the most advanced general-purpose LLMs available, with demonstrated excellence in Verilog-related tasks, such as Verilog generation, as shown in prior studies [1][4][5].
>
> To further demonstrate the superiority of DeepRTL, we conduct experiments comparing it with models specifically trained on Verilog. We did not select Zhang et al., 2024 [6] and Chang et al., 2024b [1] for comparison, as their models are not open-sourced, and it is non-trivial to reproduce their experiments. Additionally, the reported performance in their original papers is either comparable to, and in some cases inferior to, that of GPT-3.5. In the following tables, we compare two state-of-the-art Verilog generation models, RTLCoder-Deepseek-v1.1 [7] and fine-tuned-codegen-16B-Verilog [8] with our DeepRTL-220m. Notably, RTLCoder-Deepseek-v1.1 is fine-tuned on DeepSeek-coder-6.7b, and fine-tuned-codegen-16B-Verilog is fine-tuned on CodeGen-multi-16B, both of which have significantly larger parameter sizes than DeepRTL-220m. Despite this, the superior performance of DeepRTL-220m further demonstrates the effectiveness of our proposed dataset and progressive training strategy. And we will incorporate these experimental results in the updated manuscript.
>
> | Understanding | BLEU-4 | ROUGE-1 | ROUGE-2 | ROUGE-L | Emb. Sim. | GPT Score |
> | --- | --- | --- | --- | --- | --- | --- |
> | RTLCoder-Deepseek-v1.1 | 1.08 | 21.83 | 4.68 | 20.30 | 0.687 | 0.561 |
> | fine-tuned-codegen-16B-Verilog | 0.09 | 6.54 | 0.35 | 6.08 | 0.505 | 0.311 |
> | DeepRTL-220m | 18.66 | 47.69 | 29.49 | 44.02 | 0.837 | 0.705 |
>
> | Generation (syntax) | Success Rate | Pass@1 | Pass@5 |
> | --- | --- | --- | --- |
> | RTLCoder-Deepseek-v1.1 | 48.39% | 41.94% | 77.42% |
> | fine-tuned-codegen-16B-Verilog | 50.97% | 48.39% | 70.97% |
> | DeepRTL-220m | 78.06% | 70.97% | 80.65% |
>
> | Generation (function) | Success Rate | Pass@1 | Pass@5 |
> | --- | --- | --- | --- |
> | RTLCoder-Deepseek-v1.1 | 20.00% | 16.13% | 35.48% |
> | fine-tuned-codegen-16B-Verilog | 12.26% | 9.68% | 32.26% |
> | DeepRTL-220m | 36.13% | 32.26% | 41.94% |
>
> [1] Data is all you need: Finetuning LLMs for Chip Design via an Automated design-data augmentation framework, DAC 2024
>
> [3] https://openai.com/index/introducing-openai-o1-preview/
>
> [4] Verigen: A large language model for verilog code generation. TODAES 2024.
>
> [5] RTLCoder: Fully Open-Source and Efficient LLM-Assisted RTL Code Generation Technique. TCAD 2024.
>
> [6] MG-Verilog: Multi-grained Dataset Towards Enhanced LLM-assisted Verilog Generation. LAD 2024.
>
> [7] https://huggingface.co/ishorn5/RTLCoder-Deepseek-v1.1
>
> [8] https://huggingface.co/shailja/fine-tuned-codegen-16B-Verilog

---

### Official Review · Reviewer_hfVY · 2024-11-09

**Soundness:** 4
**Presentation:** 3
**Contribution:** 3
**Rating:** 8
**Confidence:** 2

**Summary:**

This paper proposes a dataset and a model for verilog generation and understanding. It carefully describes the annotation process for the dataset and presents an extensive battery of experimental results. Overall, the paper seems valuable to me, although I should clarify that I am well-versed in code generation, but not in Verilog so I may be missing some context with related work.

**Strengths:**

- As a whole, the work seems extensive and relatively careful, from conceptualization to base data collection, human annotation, model training, and evaluation.
- I am not an expert in EDA, but it seemed like the work was novel from the point of view of such a dataset and model not existing previously.
- The experimentation is extensive, comparing a fairly large number of models with various evaluation metrics.

**Weaknesses:**

- As someone who is not well-versed in Verilog, I would have appreciated an explanation of the basics of the language, what is its basic syntax, characteristics, etc. But there was not very much explanation in the paper.
- Conceptually, the work was rather straightforward and I did not get many clear research insights from the paper. For this paper I am not extremely concerned about this though, as the work seems valuable nonetheless, and could serve as a base for future research.
- It was not clear how much of the work will be released for the research community to build on. It seems that some of the data may be released, but presumably the proprietary data will not be? And also it wasn't clear about the model.

**Questions:**

None

---

> ### Author Response · Authors · 2024-11-21
> **Response to Reviewer hfVY (Part 1/2)**
>
> ### Q1: “An explanation of the basics of Verilog, including its basic syntax, characteristics, *etc*. ”
>
> R1: Thank you for highlighting this point. Below, we provide an overview of Verilog, including its basics, key differences from software programming languages, and the unique challenges involved in building a foundation model for Verilog.
>
> 1. **Basics of Verilog:** Verilog is the most widely used hardware description language (HDL) for modeling digital integrated circuits. It enables designers to specify both the behavioral and structural aspects of hardware systems, such as processors, controllers, and digital logic circuits. Verilog operates at a relatively low level, focusing on gates, registers, and signal assignments—each representing physical hardware components. While Verilog supports behavioral constructs (*e.g.*, `if-else`, `case`) that are somewhat similar to software programming languages, their use is constrained by synthesizable coding styles required for hardware implementation.
> 2. **Differences between Verilog and Software Programming Languages:**
> - **Parallelism:** Verilog inherently models hardware’s concurrnet nature, with multiple statements executing simultaneously. In contrast, software languages like Python typically follow a sequential execution model.
> - **Timing:** Timing is a fundamental concept in Verilog that directly influences how digital circuits are designed and simulated. Verilog relies on clocks to synchronize sequential logic behaviors, enabling the precise modeling of synthronous circuits. In contrast, software programming languages generally do not have an inherent need for explicit timing.
> - **Syntax and Constructs:** Verilog’s syntax is tailored to describe the behavior and structure of digital circuits, reflecting the parallel nature of hardware. Key constructs of Verilog include:
>     - **Modules:** The basic unit of Verilog, used to define a hardware block or component.
>     - **Always block:** Used to model sequential behavior, triggered by changes in signals or clock edges.
>     - **Sensitivity list:** In an `always` block, the sensitivity list specifies the signals that trigger the block’s execution when they change.
>     - **Assign statements:** `assign` statements are used to describe continuous assignments of signal values in parallel, reflecting the inherent concurrency of hardware.
>     - **Registers (`reg`) and Wires (`wire`)**: `reg` is used for variables that retain their value (*e.g.*, flip-flops or memory), and `wire` is used for connections that propagate values through the circuit.
>
>     In contrast, software programming languages like C, Python, or Java employ a more conventional syntax for defining algorithms, control flow, and data manipulation. These languages use constructs like loops (`for`, `while`), conditionals (`if`, `else`), and functions or methods for structuring code, with data types such as integers, strings, and floats for variable storage.
>
> 3. **Challenges:** As noted, Verilog significantly differs from software programming languages, with unique characteristics tailored to hardware design. As a result, transferring knowledge from existing software foundation models to Verilog is nontrivial. Moreover, Verilog is a low-resource language, which is underrepresented in conventional code datasets. As shown in [1], Verilog repositories contain orders of magnitude fewer files than those for general-purpose programming languages, making it difficult to gather the large datasets required for training a robust foundation model. In addition to data scarcity, the quality of existing Verilog datasets is often subpar, with weak alignment between natural language descriptions and Verilog code. This misalignment further hinders the model's ability to learn accurate mappings between textual specifications and hardware designs, which is critical for Verilog understanding and generation.
>
> We recognize that the absence of this information may cause confusion for readers who are unfamiliar with Verilog. To address this, we will revise the manuscript to include a section on the basics of Verilog.
>
> [1] Data is all you need: Finetuning LLMs for Chip Design via an Automated design-data augmentation framework, DAC 2024

---

> ### Author Response · Authors · 2024-11-21
> **Response to Reviewer hfVY (Part 2/2)**
>
> ### Q2: “Did not get many research insights from the paper” & “Could serve a base for future research”
>
> R2: Thank you for your thoughtful feedback. We appreciate your concern regarding the clarity of the research insights represented in our paper. The main contributions of our work are as follows:
>
> 1. **A High-Quality, Comprehensive Dataset:** We introduce a high-quality dataset that aligns Verilog code with rich, multi-level natural language descriptions.
> 2. **A Unified Model for Verilog Understanding and Generation:** We present the first model that bridges Verilog understanding and generation, along with a novel benchmark for Verilog understanding.
>
> In addition to these contributions, we recognize the significant impact that dataset quality has on model performance, and thus, we have designed a meticulous annotation strategy using Chain-of-Thought (CoT) to ensure a strong alignment between Verilog code and natural language across multiple levels. To fully leverage the potential of this dataset, we employ a progressive training strategy during fine-tuning. This comprehensive dataset, coupled with our progressive training approach, enables the development of DeepRTL, a unified model that excels in both Verilog understanding and generation, even with a base model containing only 220M parameters. Notably, while previous works have adopted much larger models (*e.g.*, Llama 2 7B & 13B in [1]), their performance is either inferior or only comparable to GPT-3.5, primarily due to the poor quality of the datasets. In contrast, DeepRTL’s superior performance over GPT-4 and o1-preview highlights the importance of both dataset quality and our training methodology.
>
> Additionally, we introduce two novel evaluation metrics, embedding similarity and GPT score, to assess the semantic similarity between generated descriptions and ground truth summaries. These metrics provide a more accurate reflection of model performance on code understanding tasks than traditional evaluation metrics like BLEU and ROUGE. To the best of our knowledge, this is the first time these metrics have been applied to code understanding, and we believe they provide a more robust and reliable means of evaluation.
>
> Furthermore, since we employ CodeT5+, a family of encoder-decoder code foundation LLMs, as the base model to train DeepRTL, we can naturally extract Verilog representations from the encoder component of the model. These representations are potentially applicable to various downstream tasks in Electronic Design Automation (EDA) from the RTL stage, including **PPA (Power, Performance, Area) prediction**, which estimates the power consumption, performance, and area of an RTL design, and **verification**, which ensures that the RTL design correctly implements its intended functionality and meets specification requirements. Our model, therefore, has the potential to serve as a foundation for future research in the field. In subsequent work, we plan to explore how DeepRTL can further enhance the productivity of the hardware design process.
>
> We hope these clarifications highlight the key insights and contributions of our work, and we will revise the manuscript to make these points more explicit. We are happy to provide any further clarification or engage in additional discussions regarding our findings and their implications.
>
> ### Q3: “How much of the work will be released for the research community to build on?”
>
> R3: Thank you for raising this point. We plan to release all components of our work, including the full dataset (comprising open-source and proprietary Verilog code along with their corresponding multi-level natural language descriptions), the Verilog understanding benchmark, and the model checkpoints, along with the training and evaluation scripts, soon after the paper is accpeted.
>
> [1] Data is all you need: Finetuning LLMs for Chip Design via an Automated design-data augmentation framework, DAC 2024

---

### Author Response · Authors · 2024-11-27
**Official Comment by Authors**

Dear Reviewers,

We would like to sincerely thank you for your valuable time and thoughtful feedback throughout the review and rebuttal process. Your detailed comments and constructive suggestions have been instrumental in enhancing the quality of our work.

In response to your feedback, we have revised the manuscript accordingly, incorporating the necessary changes and adding additional experiments where appropriate. We hope that the updated version could meet your expectations.

If you have any further questions or require additional clarifications, please do not hesitate to reach out. We truly appreciate your consideration of our work.

Best regards,

Authors

---

### Public Comment · ~Sandesh_Adhikary1 · 2025-12-16
**Where can I download the data?**

Is the dataset for this paper publically available? The paper says "Our code and datasets are available at https://github.com/PeterLau61/DeepRTL" but that repo is empty.

---

> ### Public Comment · ~Yi_Liu16 · 2025-12-17
> **Dataset Links Added to the GitHub Repo**
>
> Thanks for your interest and for pointing this out. We’ve now updated the GitHub repository and added the dataset download links in the README.
>
> DeepRTL fine-tuning dataset (Hugging Face): https://huggingface.co/datasets/liuyi2000/deeprtl_finetuning_dataset
>
> Please let us know if you run into any issues accessing or using the data. We’re happy to help.

---

> > ### Public Comment · ~Sandesh_Adhikary1 · 2025-12-17
> >
> > Thank you!

---

### Meta-Review · Area_Chair_gnkE · 2024-12-20

**Metareview:**

The paper addressed the understanding and generation of Verilog programming language, for which the authors created a benchmark dataset and presented a system based on fine-tuning CodeT5+.

Reviewers generally acknowledge that the created dataset and system could be a contribution of the paper. Reviewers (and myself), however, generally do not understand the nature of Verilog. To me, it's unclear whether the paper presents significant contributions. First, the Verilog language is less known and is unclear whether it will attract broad interest. Second, the presented approach is generic and provides little insight.

**Additional Comments On Reviewer Discussion:**

Reviewers generally acknowledge that the created dataset and system could be a contribution of the paper. Reviewers (and myself), however, generally do not understand the nature of Verilog. To me, it's unclear whether the paper presents significant contributions. First, the Verilog language is less known and is unclear whether it will attract broad interest. Second, the presented approach is generic and provides little insight.

---

### Decision · Program_Chairs · 2025-01-22

Accept (Spotlight)